



**Bayesian age-depth modelling applied to varve and radiometric**
**dating to optimize the transfer of an existing high-resolution**
**chronology to a new composite sediment profile from Holzmaar**
**(West-Eifel Volcanic Field, Germany)**
Stella Birlo[1*], Wojciech Tylmann[2], Bernd Zolitschka[1]
1 - University of Bremen, Institute of Geography, GEOPOLAR, Bremen, Germany
2 - University of Gdańsk, Faculty of Oceanography and Geography, Gdańsk, Poland
*Corresponding author: sbirlo@uni-bremen.de

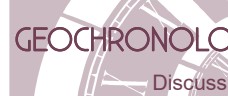 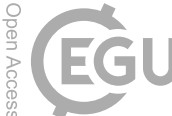

## 9 Abstract

This study gives an overview of different varve integration methods with Bacon. These techniques
will become important for the future as technologies evolve with more sites being revisited for
the application of new and high-resolution scanning methods. Thus, the transfer of existing
chronologies will become necessary, because the recounting of varves will be too time consuming
and expensive to be funded.
We introduce new sediment cores from Holzmaar (West-Eifel Volcanic Field, Germany), a volcanic
maar lake with a well-studied varved record. Four different age-depth models (A-D) have been
calculated for the new composite sediment profile (HZM19) using Bayesian statistics with Bacon.
All models incorporate new Pb-210 and Cs-137 dates for the top of the record, the latest
calibration curve (IntCal20) for radiocarbon ages as well as the new age estimation for the Laacher
See Tephra. Model A is based on previously published radiocarbon measurements only, while
Models B-D integrate the previously published varve chronology (VT-99) with different
approaches. Model B rests upon radiocarbon data, while parameter settings are obtained from
sedimentation rates derived from VT-99. Model C is based on radiocarbon dates and on VT-99 as
several normal-distributed tie-points, while Model D is segmented into four sections: Sections 1
and 3 are based on VT-99 only, whereas Sections 2 and 4 rely on Bacon age-depth models
including additional information from VT-99. In terms of accuracy, the parameter-based
integration Model B shows little improvement over the non-integrated approach, whereas the tie
point-based integration Model C reflects the complex accumulation history of Holzmaar much
better. Only the segmented and parameter-based age-integration approach of Model D adapts and
improves VT-99 by replacing sections of higher counting errors with Bayesian modelling of
radiocarbon ages and thus efficiently makes available the best possible and precise age-depth
model for HZM19. This approach will value all ongoing and high-resolution investigations for a
better understanding of decadal-scale Holocene environmental and climatic variations.
Keywords: Lacustrine sediments, Varves, Bayesian age-depth modelling, Bacon, Radiometric
dating



## 1. Introduction

Terrestrial archives from lakes have the potential to provide information about climate and the human history of its catchment area beyond instrumental and historical data. In the late 1980s, piston coring and freeze coring techniques for lacustrine sediment records have improved tremendously allowing a better quality of sediments to be recovered from modern lakes. Since then, the new fields of limnogeology and paleolimnology flourished with increasing demand of societies for documentation of natural background data related to questions around acid rain, environmental pollution and more and more with a focus on global climate change.

To provide such information not only on local scales but also on larger regional to global scales, investigations from different sites need to be compared and linked. However, such correlations are only successful if the contributing archives are based on robust chronologies. Therefore, precise and reliable age-depth models are the basis for sedimentary investigations and reconstructions of environmental and climatic changes of the past, as they ensure intra-site comparability and enable recognition of larger scale patterns. A reliable chronology should be based on a combination of different dating techniques (multiple dating approach) such as radiometric dating, well-known event layers (e.g., tephrochronology), historic data (e.g., flood events) or varve counting. The term "varve" (Swedish: layer) was first introduced by De Geer (1912) for outcrops with proglacial sediments and describes finely laminated sediment structures with annual origin. The alternating pale and dark layers are driven by seasonal changes in temperature and precipitation that cause different chemical and biological processes within the lake and its catchment area. When anoxic conditions at the sediment-water-interface are given at least seasonally, i.e. no bioturbation destroys laminations, varves are preserved and provide high-resolution and precise chronologies in calendar years (Zolitschka et al., 2015).

Until the 1980s, varve chronologies were the only option for calendar-year chronologies for sediment records, while AMS radiocarbon dating was still in its infancy and calibration of radiocarbon ages was restricted to the Middle and Late Holocene, if at all applied. First reviews about methodological advances in the study of annually laminated sediments appeared at the same time (Anderson and Dean, 1988; O'Sullivan, 1983; Saarnisto, 1986) and long varve-dated reconstructions were published for Elk Lake, USA (Dean et al., 1984) and Lake Valkiajärvi, Finland (Saarnisto, 1985). Meerfelder Maar and Holzmaar were the first varve-dated lacustrine records covering the entire Holocene and the Late Glacial for Central Europe (Zolitschka, 1989, 1988), followed by records concentrating on the Late Glacial to Holocene transition at Soppensee, Switzerland (Lotter, 1991) and at Lake Gosciaz, Poland (Goslar et al., 1993). As such, the Holzmaar record became one of the best studied lacustrine records in Europe, if not world-wide. Since the first coring campaign in 1984, several sediment records have been recovered from Holzmaar and





numerous studies were carried out with sedimentological, biological, geochemical and
geophysical methods (e.g. Zolitschka, 1989; Lottermoser et al., 1993; Hajdas et al., 1995;
Raubitschek et al., 1999; Leroy et al., 2000). However, the early sediment records from Holzmaar,
although counted and corrected multiple times, still contain sections of high counting uncertainty
and thus suffer from optimal core correlation as it is possible today by applying high-resolution
scanning techniques and digital line-scan images. Moreover, independent time control of varve
chronologies with AMS radiocarbon dating became available only in the 1990s (Hajdas-
Skowronek, 1993), while Bayesian age-depth modelling established as a tool for optimizing dating
efforts only during the last decade (Ramsey, 2009) and sediment scanning revolutionized
limnogeology and paleolimnology over the last 20 years. Therefore, we revisited Holzmaar to
obtain fresh sediment cores for the conduction of innovative and high-resolution (sub-millimetre-
scale) sediment scanning techniques to be based on an improved age-depth model.
As chronologies are always a "running target", especially as new scientific methods and
approaches appear, it is no wonder that the varve chronology for Holzmaar sediments has
developed from its first attempt as "Varve Time 1990" (VT-90) (Zolitschka, 1990) to VT-99 ten
years later (Zolitschka et al., 2000). In the course of applying ultra-high (sub-mm-scaled)
resolution scanning techniques to the new set of sediment cores from Holzmaar (HZM19), VT-99
was transferred to HZM19 making use of marker layers and radiocarbon ages for correlation as
well as of Bayesian age-depth modelling for the creation of an updated varve chronology (VT-22).
Different to earlier studies, we make use of available radiocarbon dates from Holzmaar not only
to correct the varve chronology but to combine it with the independent radiocarbon chronology
using Bayesian modelling. This integration approach is not commonly used for lacustrine records.
Here we select three different methods to integrate varve and radiometric dating and apply it to
the Holzmaar data. We concentrate on approaches using the Bacon package for the R statistical
programming software (Blaauw and Christen, 2011), whereas literature also provides
comparable methods for alternative Bayesian age-depth modelling software, such as OxCal
(Martin-Puertas et al., 2021; Ramsey, 2008; Vandergoes et al., 2018), which was also used to
integrate varve counting and radiometric dating for the Holocene sediment record HZM96-4a,4b
from Holzmaar (Prasad and Baier, 2014).
In this study we discuss the possibilities to integrate and improve different chronologies by
combining a varve chronology with modelling approaches. This is accomplished by testing and
comparing integration methods with regard to accuracy and precision from the interpolated varve
chronology itself and for a Bayesian model without any varve information. With this integration
of all age information we produce the most reliable age estimations for the HZM19 record: VT-22.
Based on the best model outcome, this master chronology serves as the chronological base for





ongoing and future biological, geochemical and geophysical investigations conducted on the new
Holzmaar sediment cores (e.g. García et al., 2022).

## 2. Materials and Methods

### 2.1    Regional Setting

The late Quaternary volcanic maar lake Holzmaar (425 m a.s.l., 50°7'8'' N, 6°52' 45'' E) is located
in the western central part of the Rhenish Massif in the West-Eifel Volcanic Field (WEVF;
Rhineland-Palatinate, Germany, Fig. 1). The WEVF consists of more than hundred volcanic cones
and maars, of which only nine are water-filled today (Meyer, 2013; Schmincke, 2014). The
volcanism in the Eifel region was caused by uplift of the Rhenish Shield since 700 - 800 ka, which
started in the NW near Ormont (Meyer and Stets, 2002; Schmincke, 2007). Volcanic activities
reached a peak at ca 600 – 450 ka in the central WEVF and then decreased towards Bad Bertrich
in the SE (Schmincke, 2007). The uplift is responsible for many eruptive centres at NW-SE
trending tectonic faults, along which several phreatomagmatic maar explosions occurred (Büchel,
1993; Lorenz, 1984; Lorenz et al., 2020; Meyer, 1985). One of these eruptions formed the
Holzmaar system ca. 40 - 70 ka ago (Büchel, 1993) consisting of three maars with the maar lake
of Holzmaar, the raised bog of Dürres Maar and the dry Hetsche or Hitsche Maar (from SE to NW).
With 100 m in diameter, the latter is the smallest maar of the WEVF (Fig. 1).
The catchment area of Holzmaar (2.06 km$^2$) includes the Sammetbach, a creek that flows in and
out of the lake. Due to the low erosive energy of the stream no delta formed in the lake (Scharf,
1987; Zolitschka, 1998a). The geology in the catchment area consists of Devonian metamorphic
slates, greywackes and quartzites as well as Quaternary loess and volcanic rocks related to
eruptions of the Holzmaar system (Meyer, 2013). Holzmaar is located within a conservation area
since 1975 protecting the surrounding beech forest (F*agus sylvatica* L.), while ca. 60% of the
catchment area is in agricultural use (Kienel et al., 2005).
Holzmaar has a diameter of 300 m (water surface: 58,000 m$^2$) and with a maximum water depth
of 19-20 m shows a deep and steep-sided morphology typical for maar lakes. Only a small and
shallow embayment in the SW interrupts the nearly circular and 1100 m long shoreline. This
appendix-like bay developed due to an artificial damming in the late Middle Ages, which was
constructed to supply a downstream water mill (Zolitschka, 1998a). For the last glacial,
paleolimnological investigations indicate oligotrophic conditions, but eutrophication started
already at the onset of the Late Glacial (García et al., 2022). During the Holocene, water quality is
affected by human activities, which started during the Neolithic (around 6500 cal. BP) according
to pollen analysis (Litt et al., 2009). Together with the inflow of the Sammetbach this caused a
steady but slow process of eutrophication and today leads to meso- to eutrophic conditions (Lücke



140 et al., 2003; Scharf and Oehms, 1992; Zolitschka, 1990). The lake is holo- and dimictic with an

141 anoxic hypolimnion during summer stratification (Scharf and Oehms, 1992). Altogether, this

142 caused a high potential for varves to be formed and preserved.

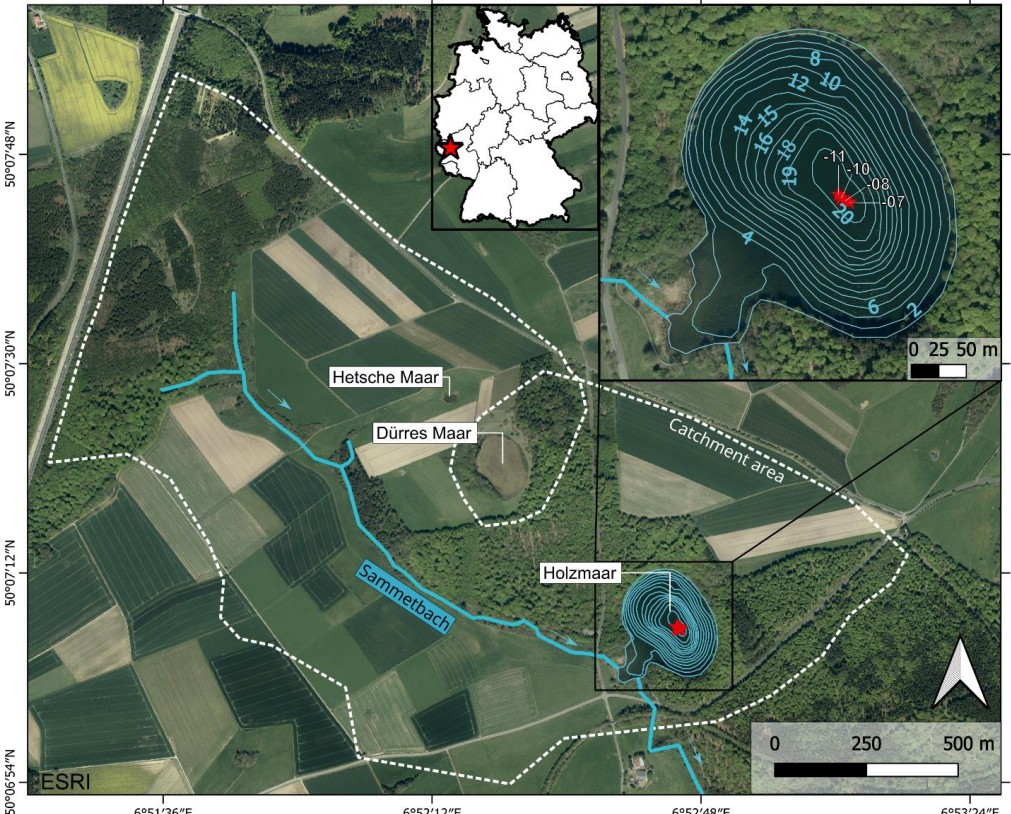


144 *Figure 1: ESRI Satellite image of the Holzmaar volcanic system and its catchment area (indicated by a white dashed line)*

145 *with Holzmaar, Dürres Maar, Hetsche Maar and Sammetbach (blue line, flow direction indicated by arrows). Upper left*

146 *insert: Location of Holzmaar in Germany (red star). Upper right insert: Bathymetric map with isobaths in meter and coring*

147 *locations (HZM19-07, -08, -10 and -11) marked by red stars.*

### 148  2.2 Sediment core collection

149 In August 2019 Holzmaar was revisited and four parallel cores (HZM19-07, HZM19-08, HZM19-

150 10, HZM19-11) have been retrieved from the centre of the lake in 19 m water depth (Fig. 1) using

151 a UWITEC piston-corer with a diameter of 90 mm (HZM19-07, -08, -10) and 60 mm (HZM19-11)

152 from a coring platform. The water-sediment interface was perfectly recovered with HZM19-07-

153 01 as the piston stopped 15 cm above the sediment surface. At the GEOPOLAR lab (University of

154 Bremen) the cores have been split in halves lengthwise, photographed and visually described

155 using a Munsell colour chart and according to the description guide line by Schnurrenberger et al.





(2003). Cross correlation of all sediment-core sections was conducted macroscopically using
distinct layers.

## 2.3   Chronology

### 2.3.1 Evolution of the Holzmaar varve chronology

First varve counts and documentation of the annual origin for the finely laminated sediments
preserved in the Holzmaar record were carried out in the late 1980's (Zolitschka, 1990, 1991,
1992), presenting the initial Holocene and Late Glacial varve chronology VT-90. Varve Time (VT)
refers to varve (calendar) years before 1950 CE (Common Era), which is equivalent to the
commonly used reference timescale for radiocarbon dates provided in cal. BP (calibrated years
before present, i.e. 1950 CE). The chronology of VT-90 was elaborated for the HZM84-B/C
composite record recovered in 1984 and was counted back to the onset of the Late Glacial, i.e. to
12,794 VT-90. This varve chronology was subsequently extended by counting the deeper,
periglacial section back to the Last Glacial Maximum, i.e. to an age of 22,500 VT-90 (Brauer, 1994;
Brauer et al., 1994).
By including the new sediment cores of HZM90-E/-F/-H, VT-90 was modified resulting in VT-94.
These overlapping sediment-core series as well as all other mentioned cores have been recovered
from the deepest part of Holzmaar, i.e. from within the 20-m isobath (Fig. 1). This recounting
revealed an underestimation of the youngest 5000 years, for which 555 years have been added.
This initial underestimation was mainly caused by sections with very thin varves difficult to count
(Zolitschka, 1998b). Another discrepancy occurred within the sediments of the Younger Dryas
(YD), for which 245 years had to be added. Altogether, the difference from VT-90 to VT-94
comprises an addition of 800 years, shifting the basal age of the Late Glacial back to 13,594 VT-94
(Zolitschka, 1998b).
To crosscheck the varve chronology with an independent dating method, 41 samples of terrestrial
macrofossils along the entire profile (Tab. A2) have been analysed using the AMS (Accelerator
Mass Spectroscopy) radiocarbon method (Hajdas et al., 1995 and one unpublished radiocarbon
date). A comparison between VT-94 and the calibrated radiocarbon chronology shows a
discrepancy of +346 years between 3500 and 4500 VT-94 (Hajdas et al., 1995; Hajdas-Skowronek,
1993). This correction factor was estimated by Chi$^2$-minimization and added by linear
interpolation between 3500 and 4500 VT-94. The outcome was VT-95, which consists of three
segments. Segment I is covered by an "absolute" chronology until 3500 VT-95, while segment II
(3500 - 4846 VT-95) was extended based on the discrepancy detected between varve and
calibrated radiocarbon chronologies. Segment III covers sediments from 4846 – 13,940 VT-95 and
is considered as a floating chronology (Hajdas et al., 1995; Zolitschka, 1998b).





In 1996 new sediment cores (HZM96-4a, -4b) have been obtained from Holzmaar and VT-95 was
transferred to this new record using 26 distinct marker layers with their related VT and error. The
age-depth model was subsequently obtained by linear interpolation (Baier et al., 2004). At the
same time, novel varve counts for the Meerfelder Maar sediment record established 1880 varve
years between the two isochrones of Laacher See Tephra (LST, eruption ca 40 km NE from
Holzmaar) and Ulmener Maar Tephra (UMT, eruption ca 13 km NE from Holzmaar) (Brauer et al.,
1999), which both are also archived in the Holzmaar sediment record. However, this well-
constrained time interval was only 1560 years long for the Holzmaar record. The obviously
missing 320 years have been positioned and added to VT-95 based on pollen data from Holzmaar
(Leroy et al., 2000), assuming a hiatus for the middle part of the YD biozone at 12,025 VT-95. This
results in the latest version (VT-99) of the Holzmaar varve chronology (Zolitschka et al., 2000)
with a basal age of 14,260 VT-99 for the Late Glacial.
Varve quality and error estimations were first discussed and described based on multiple counts
of selected and representative thin sections (Zolitschka, 1991). Later, different varve quality
classes have been described in more detail for VT-90 (Zolitschka et al., 1992) and for VT-95
(Zolitschka, 1998b) with error estimations in the 1σ range (Table A1). Similar error margins were
confirmed by counting more recent sediment profiles (HZM96-4a, 4b) from Holzmaar (Prasad and
Baier, 2014). In this study, the uppermost part was discussed as showing even higher counting
uncertainties. However, no alternative error margins were provided for this section. Thus, we use
the data of Table A1 for further evaluations.

### 2.3.2 Transfer of VT-99 to HZM19

The varve chronology VT-99 (Zolitschka et al., 2000) was transferred to HZM19 by using 43
predefined marker layers and 41 radiocarbon sampling positions analysed by Hajdas et al. (1995,
2000) with their specific VT-99 ages and errors (Tables A1, A2). Both, marker layers and
radiocarbon sampling positions have been identified and justified by comparison with documents
describing the samples as well as core photographies from previous studies and sediment profiles,
such as HZM90-E, -F, -H and HZM96-4a, 4b. All marker layers cover an age range from 141 to
14,158 VT-99. After assignment, the ages of the marker layers have been linearly interpolated and
cumulative counting errors were calculated based on the 1σ errors provided with Table A1.

### 2.3.3 Pb-210 and Cs-137 dating

The isotopes Pb-210 and Cs-137 have been used to radiometrically date the uppermost part of
HZM19 at the University of Gdansk. In total, 61 samples were taken with a thickness of 2 cm. The
activity of Cs-137 was determined directly by gamma-ray spectrometry from freeze-dried and
homogenized samples. Gamma measurements were carried out using a HPGe well-type detector
(GCW 2021) with a relative efficiency of 27% and full width at half maximum (FWHM) of 1.9 at



the energy of 1333 keV (Canberra). Energy and efficiency calibration were done using reference
material CBSS-2 (Eurostandard CZ) in the same measurement geometry like the samples. The
counting time for each sediment sample was 24 hours.
Activity of total Pb-210 was determined indirectly by measuring Po-210 using alpha
spectrometry. Dry and homogenized sediment samples of 0.2 g were spiked with a Po-209 yield
tracer and digested with concentrated $HNO_3$, $HClO_4$ and HF at a temperature of 100 °C using a CEM
Mars 6 microwave digestion system. The solution obtained was evaporated with 6M HCl to
dryness and then dissolved in 0.5M HCl. Polonium isotopes were spontaneously deposited within
four hours on silver discs. Activities were measured using a 7200-04 APEX Alpha Analyst
integrated alpha-spectroscopy system (Canberra) equipped with PIPS A450-18AM detectors.
Samples were counted for 24 hours. A certified mixed alpha source (U-234, U-238, Pu-239 and
Am-241; SRS 73833-121, Analytics, Atlanta, USA) was used to check the detector counting
efficiencies.

### 238  2.3.4 Bayesian age-depth modelling

To produce the chronology for HZM19 we test and compare different methods integrating varve
counts with radiometric measurements using Bayesian age-depth modelling. The advantage of
any modelling approach is that all possible calendar ages of calibrated radiocarbon dates and their
probability density functions (PDFs) will be tested by using a repeated random sampling method
(Blaauw, 2010; Telford et al., 2004). In addition, using the Bayes theorem allows to incorporate
information of the accumulation history known prior to modelling. Thus, calendar ages, which are
monotonic with depth and with positive accumulation rates in yr cm$^{-1}$ (in sedimentological terms,
accumulation rates as they are used for Bayesian age-depth modelling are equivalent to
"sedimentation rates", as corroborated by the units used) are calculated (Lacourse and Gajewski,
2020; Trachsel and Telford, 2017). This is different if compared to the "CLassical Age-depth
Modelling" carried out by CLAM (Blaauw, 2010).
Currently established programs that use Bayesian statistics are Oxcal (Ramsey, 2008), BChron
(Haslett and Parnell, 2008) and Bacon (Blaauw and Christen, 2011), all of which differ in terms of
parameter settings and handling of outliers. In this study, we focus on varve counting integration
methods using Bacon (rBacon version 2.5.7; Blaauw et al., 2021; Blaauw and Christen, 2011) for
the R programming language (version 4.1.1; R Core Team, 2021). Bacon uses a Markov Chain
Monte Carlo (MCMC) sampling strategy to model the accumulation history piecewise using a
gamma autoregressive semi-parametric model (Blaauw and Christen, 2011). The accumulation
rate of each segment depends on the accumulation rate of the previous segment. Dates are treated
using a student's t-distribution. Although Bacon provides default values, the accumulation rate is
controlled by two adjustable prior distributions (prior model), the accumulation rate as a gamma



distribution and the memory, which describes the dependence of accumulation rates between
neighbouring depths as a beta distribution. Both latter parameters are defined by a shape and a
strength prior, respectively, in addition to a mean prior. Furthermore, we make use of the number
of segments (thick-parameter) recommended by Bacon. The program also allows to incorporate
information about hiati and slump events in the profile.
Only few studies use the Bayesian approach that integrates varve counting information with
radiocarbon dates. We extracted three different methods and for comparison include one model
only with radiocarbon data, i.e. excluding any VT-99 information. Thus, four different age-depth
models (A-D) are compared and discussed:
A)  Model based only on radiocarbon dates.
B) This parameter-based varve integration method introduced by Vandergoes et al. (2018)
compares several varve integration techniques for sediments from Lake Ohau (New Zealand)
using both OxCal and Bacon. Here, we select the integration approach with Bacon, where the
"varve counts function" is the source for the prior-parameter of mean accumulation rate. Major
changes in accumulation history recorded by the varve data are derived by using the R package
"segmented" (Muggeo, 2022). It dissects the sediment sequence and for each resulting segment
an individual mean accumulation-rate prior is defined.
C) The tie point-based integration used by Shanahan et al. (2012) integrates the varve chronology
from Lake Bosumtwi (Ghana) based on certain tie points with normally distributed age
uncertainties of the cumulative error. They address the problem of integrating all individual varve
counts, as they cannot be considered as independent chronological datapoints. Thus, they would
be weighted too strongly in the model.  The compromise we have chosen for this study, is placing
one varve tie-point every 100 years. As there is no varve counting available for HZM19 but VT-99
ages based on marker layers, we implement them with cumulative errors as tie points instead.
D) The segmented and parameter-based integration introduced by Bonk et al. (2021) provides
the most complex method for varve integration. The problem of not or poorly varved sections in
the sediment profile of Lake Gosciaz (Poland) is compensated by dividing the profile into three
sections and interpolating the section with low-quality varves using Bayesian modelling. For the
Holzmaar record, we define four sections: sections 2 and 4 are based on Bayesian modelling, while
sections 1 and 3 rely on VT-99. Section 3 is treated as a floating chronology and placed based on
the sum of calibrated radiocarbon probabilities lying within this section. To tighten the two
Bayesian modelled sections to the following varved sections, an anchor tie-point based on the
oldest age of the younger sections is implemented.





For each model we use the calibration curve IntCal20 (Reimer et al., 2020) and make use of the
default accumulation strength and memory priors. We also implement a surface age of -69 +- 1
cal. BP as tie point with a normal distributed error to anchor the chronology to present-day.

# 3. Results and Interpretation

## 3.1  Lithology

The four parallel cores HZM19-07, -08, -10 and -11 were aligned and correlated to form the
composite profile HZM19 (Fig. 2), which includes 24 core sections and reaches to a basal depth of
14.64 m (Table A3). One technical sediment gap exists at a composite depth of 10.90 m. To
determine the precise length of this gap, we use core photographies from a previous Holzmaar
core (HZM90-H5u) and determined the technical gap with a length of 12.9 cm (Fig. A1).
The lithological description of HZM19 follows the characterization of Zolitschka (1998a, 1998b),
dividing the HZM84-B/C profile into 12 lithozones (H1 – H12). Except H1, all lithozones cover
finely-laminated diatomaceous gyttja with varying minerogenic and organic content and colour.
All lithozone depths are summarized in Table A4. The transition from light greenish grey (10Y
8/1) and greyish brown (2.5Y 5/2) minerogenic, finely laminated, weakly carbonaceous silts and
clays in H1 (12.9 – 14.6 m) to carbonaceous laminated gyttja in light olive brown (2.5Y 5/3), black
(10YR 2/1) and light-yellow brown (2.5Y 6/3) with slightly higher organic content in H2 (11.3 –
12.9 m) indicates the transition from the Pleniglacial to the Late Glacial.
Within H2, the distinct and almost 20 cm thick coarse-grained tephra from the Laacher See
eruption (LST, 11.5 – 11.7 m) is deposited, a well-dated isochrone (Reinig et al., 2021) of European
lake sediments.  The following lithozone H3 (10.9 – 11.3 m) shows a high minerogenic content
and almost no organic components with colours of light greenish grey (5GY 7/1) and grey brown
(10YR 5/2), representing the YD at the end of the Pleistocene. Unfortunately, almost one third
(12.9 cm) of the YD lithozone H3 is missing due to a technical gap.

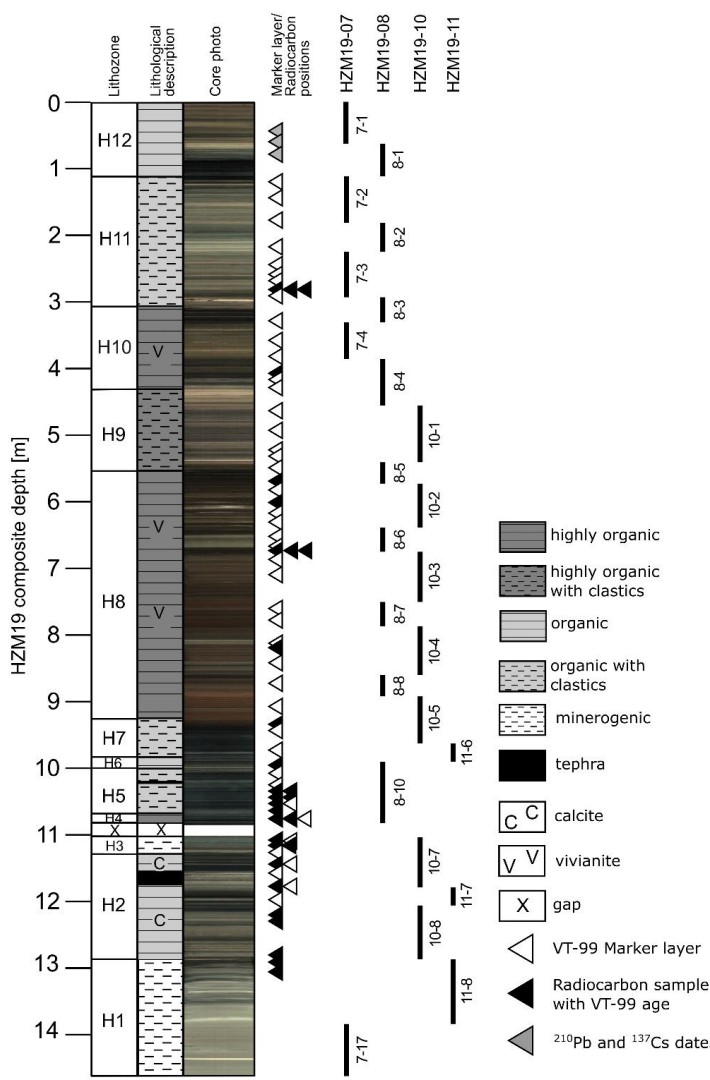


*Figure 2: Composite profile of HZM19 with (from left to right) lithozones H1 to H12 (cf., Table A4), lithological description,*

*core photographies taken immediately after core splitting, positions of marker layers and radiometric samples (cf., Tables*

*A2, A5) and core sections used for the composite profile (cf., Table A3).*

The Holocene sediment shows a periodic change from sections with higher organic content in

black (2.5Y 2.5/1) and light olive brown (2.5Y 5/3) (H4: 10.7 – 10.9 m, H6: 9.9 – 10.0 m) to sections

with high organic and clastic content in slightly brighter colours like grey (10YR 5/1) (H5: 10.0 –

10.7 m, H7: 9.3 – 9.9 m). The tephra of the Ulmener Maar eruption (UMT, ca. 3 mm thick) occurs

in H5 at 10.24 m. The longest lithozone H8 (5.5 – 9.3 m) contains distinctly varved dark reddish

brown (5YR 3/2) sediments with high organic content changing towards the top to very dark





greyish brown (10YR 3/2) and brown (10YR 4/3) with several up to 5 mm thick lenses of
authigenic vivianite. Also, a low carbonate content was recognized. Furthermore, turbidites are
observed more frequently from H8 to the top of HZM19.
Above H8, the clastic content increases and brightens up to light olive brown (2.5 Y 5/3) and
greyish brown (2.5Y 5/2) hues in H9 (4.3 – 5.5 m). In H10 (3.1 – 4.3 m) colours change to darker
hues, e.g. olive grey (5Y 4/2) and black (5Y 2.5/2), while the organic content remains high and
terrestrial macrofossils like pieces of wood or leave remains occur more frequently towards the
top. The organic content is decreasing slightly in H11 (1.1 – 3.1 m), which also contains clastic
components and terrestrial plant material as well as turbidites with paler colours, e.g. olive brown
(2.5Y 5/3) and grey (2.5Y 5/1). The uppermost lithozone H12 (1.1 m to the top of HZM19) shows
unconsolidated organic sediment with a homogenous blackish (5Y 2.5/1) colour for the lower
part and brighter dark olive grey (5Y 3/2) sediment at the very top.

## 3.2   Chronology

### 3.2.1 Pb-210 and Cs-137 dating

The profile of unsupported Pb-210 activity concentration shows a gradual rather than an
exponential decrease within the first meter of HZM19 (Fig. 3). Additionally, a plateau from 8 to
30 cm is interpreted as a section with rapid deposition of homogenous material and will be
treated for further analyses as a slump event. Despite this irregularity, the gradual decrease in
unsupported Pb-210 activity with depth indicates high sedimentation rates. We use the CFCS
(Constant Flux Constant Sedimentation) model to estimate mean sedimentation rates of
$1.09 \pm 0.13$ cm yr$^{-1}$. This value should be treated with caution but suggests that the uppermost
meter (including a 22 cm-thick slump) was deposited in ca. 70 years.

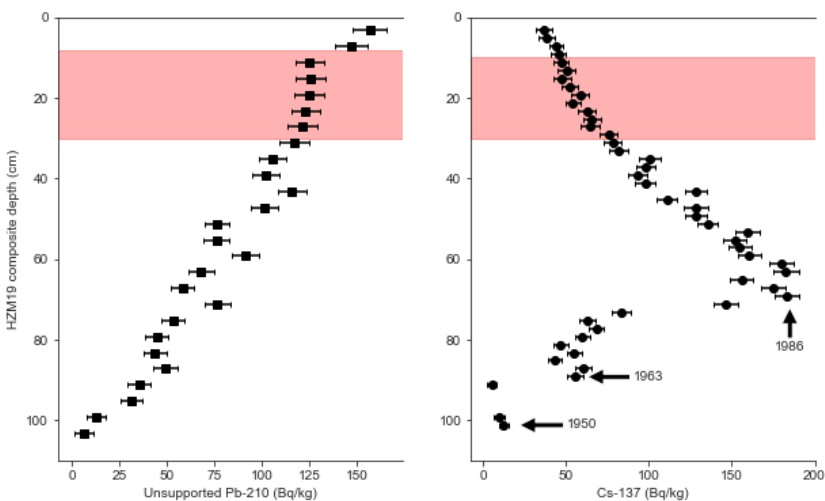


*Figure 3: Results of unsupported Pb-210 (left) and Cs-137 (right) measurements with error bars for the uppermost 110 cm*
*of HZM19. Shaded areas indicate the plateau shown by Pb-210 data, black arrows mark peaks assigned to*
*radiochronological events (given numbers are ages in years CE).*

The variability of Cs-137 activity concentrations delivers potentially three historical markers
(Fig. 3). The Cs-137 profile is smooth lacking sharp peaks due to high sedimentation rates and
likely sediment focusing. First traces of Cs-137 are recognizable at 101.2 cm and indicate atomic
bomb testing in the early 1950's. At 89.2 cm, there is a significant increase signalling
atmospheric fallout in the early 1960's in response to peak atomic bomb testing. Finally, at 69.2
cm a strong increase in Cs-137 documents the 1986 Chernobyl accident (Fig. 3, Table A5). This
interpretation is generally in line with the results of Pb-210 dating. The shape of the Cs-137
record also corresponds nicely to the results of Sirocko et al. (2013), who measured Cs-137 on
sediments from Schalkenmehrener Maar and Ulmener Maar (both WEVF). For both of these
cases, the 1986 Chernobyl peak is also much larger than the one related to the start of atomic
bomb tests in 1963.

### 3.2.2 Varve time and independent radiocarbon chronology

The varve chronology VT-99 was transferred to HZM19 using 84 marker layers of which 41 are
radiocarbon dating positions. These marker layers distribute in HZM19 from 1.16 - 12.93 m and
cover the VT-99 age range from 141 to 14,158 VT-99 (Table A2). During the transfer of marker
layers to HZM19 and comparison between HZM19 and previous Holzmaar sediment cores
(HZM84-B/C, HZM92-E/-F/-H, HZM96-4a/4b) differences in position of the lowermost marker





layers occurred. All records show differences in distances between marker layers (ML) 1 (14,156
VT-99), ML-2 (14,152 VT-99) and ML-3 (13,646 VT-99) making a clear assignment of these layers
difficult. Thus, we excluded these three marker layers for the transfer of VT-99 to HZM19. The
lowermost applied marker layer is therefore ML-4 with a varve age of 13,087 VT-99 at a depth of
11.86 m. Because of inconsistencies in documentation, we excluded two more VT-99 ages, i.e.
those related to the radiocarbon ages HZM-46 and HZM-10.1 (Table A2).
The marker layer density reaches a mean value of 5.5 dpm (dates per millennium) being most
frequent before 10,000 and after 6000 cal. BP (Fig. 4). We use a linear interpolation to receive an
age-depth model based only on VT-99 with a resulting accuracy of 282 years as a mean age range
and a maximum age range of 744 years (Table A6).
The radiocarbon dating density of HZM reaches an overall mean value of 2.7 dpm (Fig. 4), which
is 35% higher than the 2 dpm recommended for Bayesian modelling by Blaauw et al. (2018).
However, their distribution is uneven. Radiocarbon dates are most frequent for ages >10,000 cal.
BP with 3-7 dpm (mean: 5 dpm) (Fig. 4). A minimum density of radiocarbon dates (0-1 dpm) is
obtained from 10,000-6000 cal. BP (mean: 0.5 dpm). Therefore, a chronology based on the
available radiocarbon data within this section should be interpreted with caution. Dating density
for the uppermost 6000 years is higher and varies between 1 and 4 dpm (mean: 2.2 dpm).
When we compare VT-99 with radiocarbon ages calibrated with the latest calibration curve
IntCal20 (Reimer et al., 2020), an overall agreement with marker layers is observed. Only for the
lowermost part below approximately 10.64 m at radiocarbon sample HZM-46 (Table A2), we
observe an increasing underestimation of VT-99 in relation to IntCal20 calibrated radiocarbon
ages (Fig. A2, Table A2). This was already observed by Hajdas et al. (2000) in comparison to
Intcal98 but has not been corrected yet.






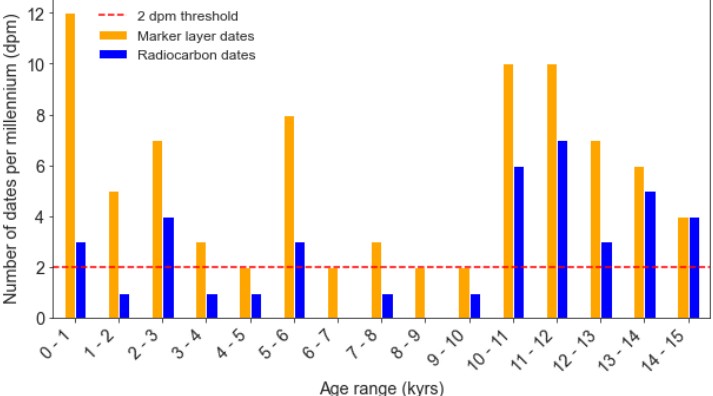


*Figure 4: Number of dating points per millennium (dpm) of HZM19 for marker layers (n: 84, mean: 5.5 dpm) and radiocarbon dates (n: 41, mean: 2.7 dpm). Red dotted line marks the recommended threshold of 2 dpm for Bayesian modelling suggested by Blaauw et al. (2018). Surface age and three ages estimated by Cs-137 are excluded.*

### 3.2.3 Age-depth modelling

Four different Bayesian age-depth models are calculated, of which three include varve ages (Model B-D) and one only radiocarbon ages (Model A). In common for all model runs are the default memory priors and the use of the IntCal20 calibration curve (Reimer et al., 2020). Furthermore, based on the Pb-210 and Cs-137 dating analysis, a slump at a composite depth of 8-30 cm was implemented, as well as the LST from 11.52 – 11.71 m. As known from previous varve and pollen studies of the Holzmaar record (Brauer et al., 1999; Leroy et al., 2000), 320 years are missing during the YD and have been included into VT-99 at 12,025 VT-99. Based on the study of Leroy et al. (2000), we were able to locate the position of the YD hiatus to a depth of 11.09 m, which we implemented for each model with a maximum duration of 320 years. In addition to marker layers and radiocarbon dates, we included the surface age of -69 +- 1 cal. BP and three events dated by Cs-137 (Table A5).

Preliminary test runs reveal two necessary changes to be made for the calculations: 1) The default number of iterations is too low to produce a robust model for the entire HZM19 sediment sequence. Thus, we use the *Baconvergance()*-function of Bacon to estimate the number of iterations needed. This function repeats the calculations and tests if the MCMC mixing of the core results in a robust model by calculating the "Gelman and Rubin Reduction Factor" (Brooks and Gelman, 1998). Good mixing is indicated by a threshold of <1.05, which in our case was reached after three iterations when the number of iterations was increased to 40,000. This results in a better mix of MCMC iterations but also in long calculation times (> 5 hours). 2) For each test run, Bacon predicted ages consistently too old for the LST, which is probably caused by slightly too old





ages of the surrounding radiocarbon dates (Table A2). To gain a better comparability with studies
from other sites, we decided to include the latest LST age of 13,006 +- 9 cal. BP (Reinig et al., 2021,
Table A5).
In addition, we extended the age-depth model to a maximum depth of 14.64 m, as ongoing
analyses exceed the lowermost dated level. However, in the following chapters we only discuss
the model output between the first (ML36/1) and the last (HZM-19) marker layer at 12.93 m
(Table A2) and compare it with the interpolated varve chronology (VT-99).
After each calculation and if the Bacon output indicates a highly variable log of objectives or MCMC
iterations, we made use of the scissor()-command to achieve a better mixing of the output. All
Bacon model outputs with their settings and additional information are shown in Fig. A3 and
related ages are listed in Table A6.
The **model without varve integration (Model A)** is based on the year of sediment recovery
(surface age), three dates estimated by Cs-137 analyses, the age for the LST (Reinig et al., 2021)
and 41 calibrated radiocarbon probability density functions (Fig. A3A). Different to Hajdas et al.
(1995), this model includes the outlier of HZM-23, but excludes HZM-24 and other described
outliers (Table A2).
Model A results in an age of 14,615 [minimum: 14,339, maximum: 14,926] cal. BP at the
lowermost dated depth of 12.93 m with a mean age uncertainty of 468 yrs. The maximum age
uncertainty of approx. 1056 years occurs at a depth of 8.86 m within lithozone H8 (Table A6),
where radiocarbon dating density is <1 dpm (Fig. 4).
The **parameter-based integration (Model B)** integrates VT-99 using all dates as in Model A and
adjusts the prior information given for the calculation based on the varve accumulation-history.
We follow the procedure presented by Vandergoes et al. (2018) and calculate a breakpoint based
on ages and depths of the marker layers at 4.43 m, i.e. at 1312 VT-99 (Fig. A3B). This boundary is
implemented as an additional hiatus to the Bacon code with a duration of 1 year. The accumulation
rate prior is set based on published sedimentation rates (Zolitschka et al., 2000). We calculate
with a mean of 0.49 yr/mm for the uppermost part (71-1312 VT-99), with 1.30 yr/mm from 1312
to the YD hiatus at 12,025 VT-99 and with 0.76 yr/mm from the YD hiatus to the lowermost age
of 14,158 VT-99. Model B is calculated using the same parameters as for Model A and with the
same treatment of outliers.
The resulting posterior model shows similarities to Model A, having a maximum mean age of
14,456 [min.: 14,236, max.: 14,749] cal. BP at a depth of 12.93 m and a mean 95% confidence
interval of 456 years with a maximum of 1064 years at 8.78 m, i.e. within the period of lowest
radiocarbon dating density (Fig. 4).



The **tie point-based integration (Model C)** is based on the approach used by Shanahan et al.
(2012). We include 43 marker layers with related VT-99 ages and cumulative errors as normal
distributed tie points into the model, which adds to the dates used in Models A and B and sums up
to 89 dates. This approach increases the amount of chronological information and fills areas with
larger gaps between radiocarbon dates. The model was run with default settings provided by
Bacon (Fig. A3C). Bacon recognizes the outliers in the same way as by previously described
models.
Model C results in a maximum age of 14,614 [min.: 14,332, max.: 14,919] cal. BP (at 12.93 m) with
a mean 95% confidence interval of 329 years, which is better than for Models A and B. A maximum
age range of 749 years is given at a depth of 9.18 m, which is also slightly better than for previously
presented models. However, Model C produced MCMC iterations with highest noise and it was
difficult to cut out a well-mixed section (Fig. A3C, upper left panel).
The **segmented and parameter-based integration (Model D)** is a more complex method of
varve integration used by Bonk et al. (2021) and was adapted for the HZM19 profile by dividing
the varve chronology of VT-99 into four sections. This separation is based on variations of
counting uncertainty, radiocarbon sampling density and an increasing offset of VT-99 to the latest
calibration curve IntCal20 (Fig. A2).
Section 1 (0 – 5.98 m) and Section 3 (6.70 – 9.90 m) are transferred and interpolated based on VT-
99 marker layers, as they are consistent with calibrated radiocarbon data (Section 1) and have
well-preserved varves with small counting errors of ±0.7% (Section 3). Section 2 (5.98 – 6.70 m)
and Section 4 (9.9 – 14.6 m) are reported as showing higher counting uncertainties (Section 2) or
increasing differences between VT-99 and the calibration curve (Section 4). Thus, we replace the
varve chronology in Sections 2 and 4 with Bayesian age-depth modelling (Fig. A3D). Section 4 also
contains very dense radiocarbon dates (Hajdas et al., 2000), which increase the predictability of
Bacon (Fig. 4).
Section 1 is based on linear interpolation for ages of the sediment surface (-69 ± 1 cal. BP), three
dates derived by Cs-137 analyses (Table A5) and 25 ages of marker layers with a basal age of 3704
± 134 cal. BP at the position of HZM-25 (Table A2).
The modelled Section 2, previously identified as a section with sedimentation rates >2.86 yr/mm
and therefore a source of high counting uncertainties and underestimation of varve ages
(Zolitschka et al., 2000), consists of five radiocarbon dates (Table A2) and the basal age of Section
1 (3704 ± 134 cal. BP) as anchor point for Section 2. To reduce the resulting gap between first and
second sections, we reduce the error estimation for the anchor point to ±70 years (+-0.5$\sigma$). As
there is a major change in sedimentation rates within this section, we calculated a boundary



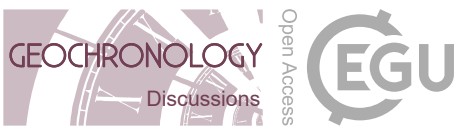

similar as in Model B using the marker layers of this section (Fig. A3D). This allows defining a boundary at the depth of 6.29 m with adjusted accumulation means of 3.33 yr/mm above (5.98 – 6.29 m) and 1.59 yr/mm below (6.29 – 6.70 m), using published sedimentation rate data (Zolitschka et al., 2000). Based on suggestions by the software, the "thick"-parameter was set to 4 mm. The resulting model covers and age range from 3709 [min.: 3591, max.: 3825] to 5419 [min.: 5329, max.: 5548] cal. BP (Fig. A3D section 2).

Section 3 interpolates 16 marker layers (Table A2), which are treated as a floating chronology. The placement of the anchor point relates to the basal age of the lowermost calibrated radiocarbon date (HZM-4.3) in Section 2 (Table A2) and the maximum sum of the four calibrated radiocarbon PDFs within this part with a summed probability of 0.076 at 5450 ± 165 cal. BP (Fig. A4 A).



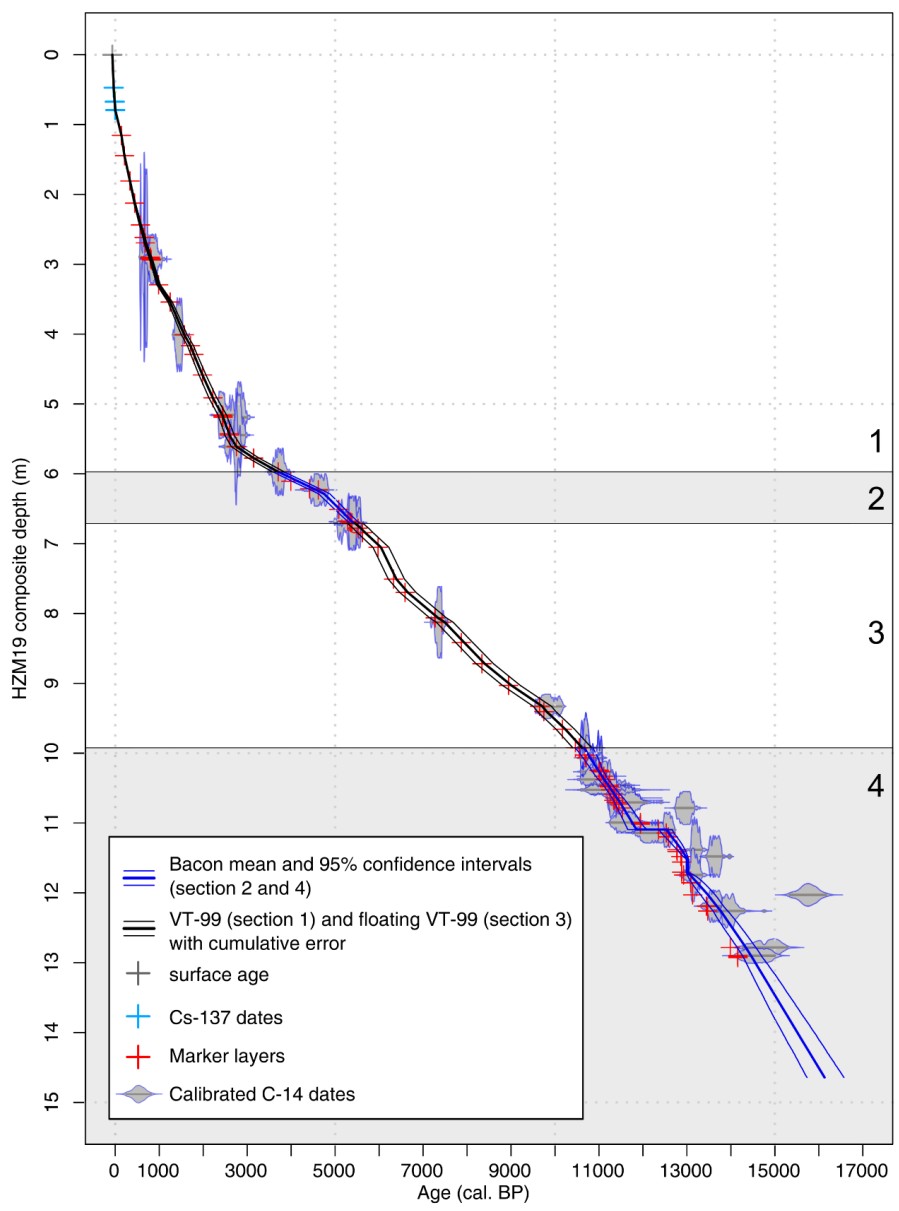


*Figure 5: Age-depth model for HZM19 based on Model D with Sections 1 and 3 based on VT-99 (section numbers at the*
*right) and Sections 2 and 4 based on Bayesian modelling (shaded).*
In comparison to the original VT-99 this approach results in a shift of +65 years for all marker
layers within Section 3 (Fig. A4 B). Thus, a basal age of 10,619 ± 213 cal. BP is obtained for Section

505 3.

The basal age of Section 3 is implemented as the anchor tie-point for the Bacon calculation of
Section 4 with a reduced error of 100 years to tighten both sections closer to each other. In



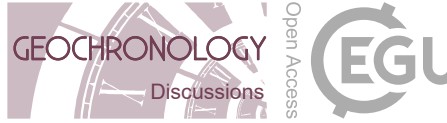

addition to the difficulties based on missing sediment within the YD, this section is the source of
highest counting uncertainties for VT-99. Section 4 is based on 25 radiocarbon dates and the latest
age estimation for the LST (Table A2). As in Section 2, we adjusted the sedimentation rate prior
(= 0.94 yr/mm) based on VT-99 accumulation rate data (Zolitschka et al., 2000). The Bacon
software suggests a segment length of 30 mm that we applied. The resulting model covers an age
range from 10,663 [min.: 10,457, max.: 10,864] to 14,485 [min.: 14,287, max.: 14,721] cal. BP at
12.93 m (Fig. A3D, Section 4).
If all sections are merged, the continuous age-depth relationship forming Model D (Fig. 5) consists
of 63% VT-99 ages and 37% Bacon modelled ages with in total 80 missing years between the
sections, as it is not possible to determine the exact start and end age of the models. This
segmented and parameter-based integration model results in a maximum age of 14,485 [min.:
14,287, max.: 14,721] cal. BP (at 12.93 m) with a mean age uncertainty of 229 years, which is the
smallest of all four tested models. The maximum age range is 447 years at 11.09 m depth and thus
considerably smaller compared to those of Models A to C (Table A6).

### 522   3.2.4 Comparison of model output with the isochrones UMT and
### 523         LST and the YD biozone

The tephra layers of UMT and LST have been identified for sediments from Holzmaar and
Meerfelder Maar (Brauer et al., 1999). The varve age of 11,000 VT-99 for UMT was derived from
the Holzmaar chronology (Zolitschka, 1998b), while the YD hiatus of this site did not allow any
calendar-year estimation for LST. As no such hiatus exists between these two isochrones at
Meerfelder Maar, the age for the LST was derived as 1880 varve years older than UMT, i.e. as
12,880 VT-99. A recent study presents a new and 126 years older age for the LST (Reinig et al.,
2021). This age of 13,006 cal. BP was implemented for the calculation of Models A-D.
When we compare all models, the age estimations for UMT and LST are close to the published ages
with the UMT dated ca. 20-50 years earlier and thus matches well within the 95% confidence
interval (Table A6). Due to the new age of LST, the distances between both isochrones vary from
2030 (Model D) to 2057 (Model C) years, which is 150-177 years more than counted for
Meerfelder Maar.
The main differences occur in prediction of the end of the YD that defines the transition to the
Holocene. The rapid cooling and subsequent warming left behind easy to recognize traces in many
European lake records increasing the comparability between sites. The entire YD is not covered
by HZM19 due to a technical gap. Nevertheless, we are able to estimate depth and time range
based on detailed pollen investigations (Leroy et al., 2000). Using VT-99, Leroy et al. (2000) date
the onset of the YD, i.e. the Allerød/Younger Dryas transition (AL/YD) to 12,606 VT-99 and the



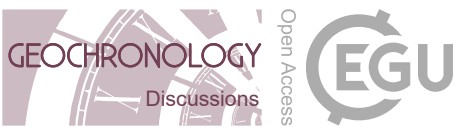

Younger Dryas/Preboreal (YD/PB) transition to 11,632 VT-99 with a 320 years hiatus at 12,025
VT-99. For HZM19 these boundaries occur at 10.88 m (YD/PB), at 11.26 m (AL/YD) and the hiatus
at 11.11 m (Fig. 6).
All model runs predict a YD duration in the range of 1012 (Model C) to 1073 (Model D) years,
which is longer than the 974 years given by VT-99 (Table A6). However, the predicted times are
closer to its duration counted for Meerfelder Maar (1080 years) (Brauer et al., 1999) or the even
longer time spans detected for Lake Gosciaz (1150 years) (Bonk et al., 2021)..
Moreover, the YD transition has been predicted within the 95% confidence interval comparable
to VT-99 (Table A6) and to the Meerfelder Maar record. Only the AL/YD transition varies between
12,694 (Model C) and 12,737 (Model B) cal. BP and, thus, is predicted earlier than for VT-99
(12,606 VT-99). However, this age range still covers the age estimations from Lake Gosciaz
(12,620 [min.: 12,389, max.: 12,753] cal. BP) and Meerfelder Maar (12,680 [min.: 12,640, max.:
12,720] cal. BP). In difference, the YD/PB transition varies between 11,655 (Model D) and 11,723
(Model B) cal. BP, which is slightly earlier than estimated by Meerfelder Maar (11,600 [min.:
11,570, max.: 11,630] cal. BP) and much earlier than the age estimation for Lake Gosciaz (11,470
[min.: 11,264, max.: 11,596] cal. BP). These discrepancies between the boundaries of the YD
biozone obtained by VT-99 and those obtained by the model runs are probably related to the new
and 126-year older age for the LST, which is included with all models. Thus, age discrepancies are
attenuating towards the UMT with 110 years at the AL/YD transition and 57 years at the YD/PB
transition.

### 3.2.5 Comparison of model output with VT-99

The comparison of all presented models differs in means and accuracies of predicted ages along
the core (Fig. 6A1; B1; C1; D1), which becomes more evident in comparison with VT-99 (Fig.
6A2,3; B2,3; C2,3; D2,3). These differences in mean modelled age and mean VT-99 age vary in
direction and amplitude (Fig. 6A2; B2; C2; D2). The largest age differences during the Holocene
occur in Model A and B with up to 300 years between 4 and 6 m depth (Fig. 6A2; B2). The defined
boundary in Model B results in large differences within the boundary area, predicting much
younger ages than VT-99. Due to the small cumulative counting uncertainty of VT-99 in the upper
part of the profile, the mean of Model B outranges the VT-99 error in most sections above 6 m (Fig.
6B2).

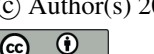

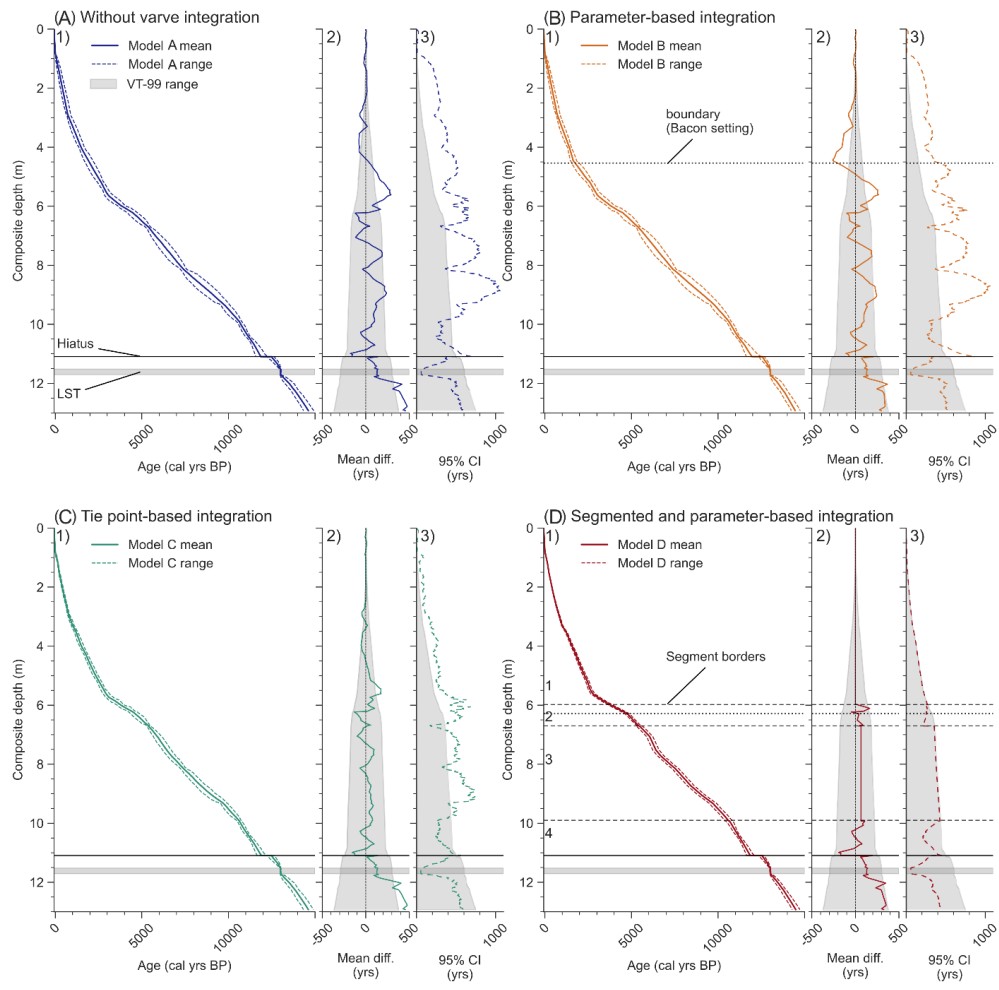

*Figure 6: Results of Models A, B, C, D plotted against composite depth (1), compared to VT-99 as the difference of mean ages (Model mean – VT-99 mean) (2) and plotted vs. VT-99 confidence intervals (CI) (3).*

The approach used for Model C reduces the difference between VT-99 and the model, probably a result of increased dating density (Fig. 6C2). This approach also leads to less over- and underestimations of the model's mean age and the VT-99 age range (Fig. 6C2). Only the segmented insertion of VT-99 in model D results in comparable ages during the Holocene (Fig. 6D2)

In the Late Glacial part below 11 m, all models produce ages constantly older than VT-99 (Fig. 6A2, B2, C2, D2). The age differences are even higher (up to 477 years), when the Bacon prior for accumulation rates was not adjusted to VT-99 (Fig. 6A2, C2). In the other cases the maximum age differences are 369 and 354 years for Model B and D, respectively (Fig. 6B2, D2). Hajdas et al. (2000) already observed a shift between the varve ages of radiocarbon dated samples and calibrated ages using the INTCAL98 calibration curve (Stuiver et al., 1998) and discuss the



difference using the LST age estimation from Meerfelder Maar (12,880 VT). However, no
adjustment has been made to fit the VT-99 ages to the calibration curve. With the LST dated to
13,006 +-9 cal. BP (Reinig et al., 2021) and the use of the INTCAL20 calibration curve, an
underestimation of VT-99 compared to the calibration curve is still existing (Fig. A2). Therefore,
a correction of ages older than 12,800 cal. BP is needed to ensure comparability of HZM19 to other
sites.
In order to find the best method to transfer VT-99 to HZM19 and to improve the chronology by
using Bayesian modelling, a closer look to each model's accuracy is necessary (Fig. 6A3, B3, C3,
D3). In comparison to the cumulative VT-99 counting error, Models A and B show maximum
differences in age uncertainties up to +655 and +665 years, respectively (Table A6). Especially
below 9.82 m, both models predict ages with larger uncertainties than the estimated counting
error for VT-99, particularly with increasing distance to radiocarbon dated levels. Therefore, no
improvement in accuracy of age estimations is observed when using the parameter-based
approach (Model B).
The tie point-based Model C also predicts larger uncertainties than VT-99 below 9.82 m (Fig. 6C),
whereas the overall difference of the age range is reduced to a mean of 47 years with a maximum
of +401 years (Table A6). Only the segmented and parameter-based Model D shows no
significantly enlarged age uncertainties and an overall improved mean age range as it adapts the
cumulative error of the varve chronology in Sections 1 and 3 (Table A6). The overall improvement
occurs in Sections 2 and 4, which is the result of more detailed prior settings for the model run.
However, all age models result in more accurate age estimations in the Late Glacial part, where
the cumulative counting error is higher and radiocarbon dating sampling is dense. But still we see
that Models C and D perform best within this section, as they predict ages with constantly lower
uncertainty ranges than VT-99. This is in contrast to the other models, which show increased and
therefore larger uncertainties at a depth of ca. 11 m. As we calculate this section in Model D with
the same data like for Model A and B, we assume that the better adjustment of the sedimentation
rate mean prior of Model D influences the model's accuracy. In terms of accuracy, there are no
general improvements in calculating a single model for the entire record, but improvements are
realised by adjusting the priors in a more detailed way.

## 4. Evaluation of the different varve integration techniques

All models predict convincing age estimations for the isochrones of LST and UMT, whereas the
prediction of the YD between both isochrones remains somewhat ambiguous, due to a
documented hiatus and too few radiocarbon ages being available for this biozone.





In terms of accuracy and precision, the varve-integration technique applied in Model D,
introduced by Bonk et al. (2021), results in most convincing age estimations for HZM19. Especially
in terms of accuracy, none of the completely Bayesian modelled age-depth relationships improved
the small age uncertainties of VT-99 in the upper part. Only in sections with markedly higher
radiocarbon sampling density or in sections with high varve counting uncertainty the Bacon
models perform better and result in more accurate age estimations than VT-99.
In comparison, Model B shows nearly no improvement over the approach without varve
integration (Model A). The reason is probably the low-resolution definition of sedimentation rate
changes (boundaries) for HZM19, which does not reflect the complex accumulation history.  Also
Vandergoes et al. (2018) reject this integration model. We suggest that this form of varve
integration is more useful for less complex and for shorter sediment profiles.
Better results are observed applying Model C, which is actually the easiest to apply. The accuracy
is improved compared to Models A and B as the dating density increases significantly. Based on
the Bayesian approach, this leads to smaller age ranges as higher uncertainties occur with
increasing distances to dated levels. The resulting mean age is more constrained by VT-99. The
accuracy might be improved by additional adjustments of the sedimentation-rate prior (here:
based on VT-99). However, varve ages inserted as tie points are included with normal distribution.
Therefore, they should not be interpreted as independent measurements with non-normal
distributed PDFs. Bayesian statistics could weight tie points too much when they are included too
densely. Therefore, this approach should be interpreted with care.
The best result in precision and especially accuracy is achieved by the segmented and parameter-
based Model D. This approach is the most challenging, but makes advantage of both, the high
accuracy of varve counting and the Bayesian approach for densely radiocarbon dated sections.
The main difference to the other models is that Model D replaces the sections of lower dating
accuracy with modelled ages that incorporate varve information and radiocarbon measurements,
which result in a much better performance.
For upcoming geochemical and geophysical studies of the HZM19 record, we will use Model D. As
parts of VT-99 (63%) are included in the new chronology, we will refer to it as chronology "VT-
22", which delivers highly accurate age estimations for each depth of the sediment profile HZM19.
Altogether, this will improve the comparability of the Holzmaar record with other sites.

## 5. Conclusion

As limnogeological and varve studies proceed, new techniques for sediment analysis develop.
Thus, previous studies can be improved by reinvestigation. However, many of the previously
studied sediment cores are not available for analysis anymore. We expect such cases to happen

more frequently in the future. Rarely, the rather time-consuming and expensive chronological studies, especially if the counting of varves is involved, will be funded a second time. This increases the need for finding best ways to adapt varve chronologies obtained during previous studies and to transfer them efficiently and precisely to new sediment cores.

For the well-dated Holzmaar record, we tested three different approaches for the integration of varve counting and radiocarbon dating using Bayesian modelling and applied them to the new composite profile from Holzmaar (HZM19). We conclude that all models result in accurate and precise age estimations. However, with higher dating density and more prior settings used to adjust the Bacon model runs, the model output is enhanced. This is confirmed by results of Model D, which improved and corrected the age estimations considerably. In contrast, Models B and C show nearly no improvement over VT-99 just like the output of Model A without varve integration.

Multiple varve counting is still one of the best approaches of building a reliable chronology for lacustrine sediment archives. However, the occurrence of hiati or errors in varve counts lead to larger uncertainties with increasing depth that need to be corrected by using independent dating techniques. Therefore, if varve and radiocarbon data are available, like it is the case for Holzmaar, the transfer of both to form a new and integrating chronology is the best option.

For this study of varve integration, we use Bacon. The parameter adjustment of Bacon is complex and especially beginners have problems to understand each single parameter and the effect it has on model results. We compare different models and settings, which helps to decide selecting the best suited approach. and to consider the parameters that have to be adjusted. Afterall, we suggest to increase the independent dating density and to adjust prior settings as detailed as possible to gain a more precise chronology for the varve-integration attempt.

Optimizing the Holzmaar chronology is the first step in order to provide a precise and robust age-depth model for upcoming and high-resolution multi-proxy investigations to unveil all the environmental details recorded by the varved sediment archive of Holzmaar.



**Appendix**

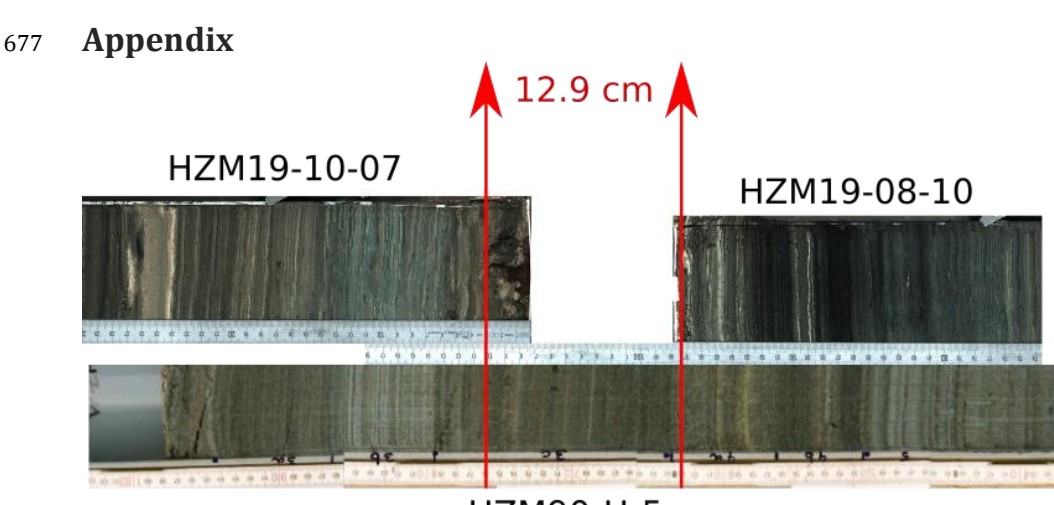


*Figure A1: Determination of the technical gap for HZM19 during the YD. This gap exists between sections HZM19-10-07*
*and HZM19-08-10 and is bridged by section HZM90-H5u from an earlier coring campaign.*



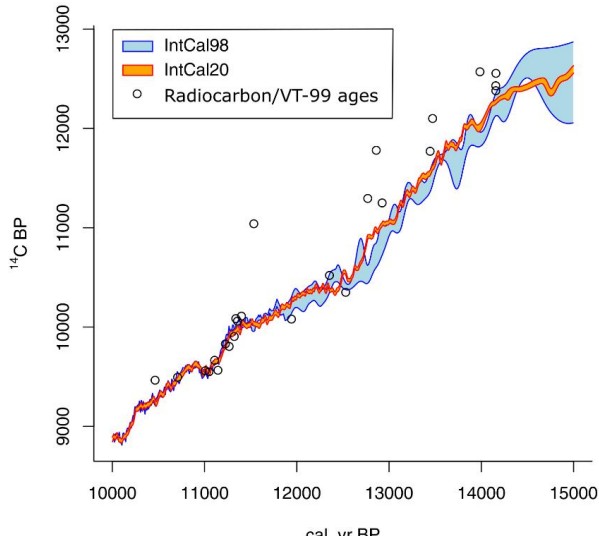


*Figure A2: Radiocarbon ages vs. Intcal98 and Intcal20 calibrated ages between 10,000 and 15,000 cal BP. Black circles
show radiocarbon ages from Holzmaar vs. VT-99 age. An underestimation of these ages occurs after 12,500 cal BP, where
VT-99 seems to be too young.*






*Figure A3: Bacon output for Model A, B, C and D (sections 2 and 4. Each output with indicator panels from top left to right:*
*MCMC iterations, prior (green) and posterior (grey) for accumulation rate distribution, memory and hiatus with defined*
*settings in red. Main panel: model with calibrated radiocarbon date probabilities (blue), tie-points with normal distribution*
*(orange) and the posterior age-depth model with mean (red dotted line) and 95% confidence intervals (gray dotted line).*
*Vertical gray lines (from left to right): slump event, defined hiatus and Laacher See Tephra. In additional panels of Models*
*B and D2 boundaries indicating major changes in accumulation rate are provided as vertical dotted lines.*



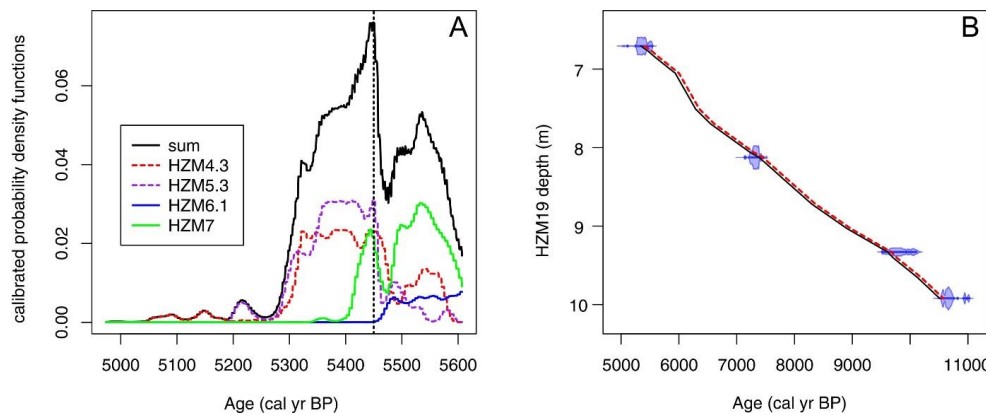


*Figure A4: Calculations for the floating VT-99 chronology of Model D, section 3. A: Calculation of shift based on calibrated*
*probability density functions of each radiocarbon sample within this section. The maximum summed probability occurs at*
*5450 cal BP. B: Original VT-99 (black) vs. floating VT-99 (+65 years, red dotted) with calibrated radiocarbon samples*
*plotted vs. depth.*



*Table A1: Error (1 sigma) estimations for different varve quality periods for the Holzmaar record (Zolitschka, 1998b),*
*updated from VT-95 to VT-99.*

| Varve quality period | VT-99 (duration in years) | Error |
| --- | --- | --- |
| A | 0 – 2800 | ±4.0 % |
| B | 2800 – 5300 | ±2.6 % |
| C | 5300 – 11,600 | ±0.7 % |
| D | 11,600 – 14,158 | ±5.9 % |
| Entire record | 0 – 14,158 | ±2.5 % |








*Table A2: Marker layers (in italics) and radiocarbon dates (Hajdas et al., 2000, 1995 plus one unpublished radiocarbon*
*date) vs. composite depth of HZM19. The calibrated median ¹⁴C age is calculated using OxCal with the IntCal20 calibration*
*curve. Inconsistent calibrated ages are shown in brackets.*

| Marker layer and ¹⁴C sample ID | HZM19 depth (m) | VT-99 Age (cal BP) | VT-99 cumulative ±1σ error (yrs) | ¹⁴C age (BP) | ¹⁴C ±1σ error (yrs) | Calibrated ¹⁴C median age (cal BP) | ¹⁴C ±1σ error (yrs) |
|---|---|---|---|---|---|---|---|
| *ML-36/1* | *1.16* | *141* | *6* | | | | |
| *ML-36* | *1.45* | *209* | *8* | | | | |
| *ML-35/1* | *1.81* | *334* | *13* | | | | |
| *ML-35* | *2.12* | *442* | *18* | | | | |
| *ML-34* | *2.44* | *572* | *23* | | | | |
| *ML-33/2* | *2.62* | *657* | *26* | | | | |
| *ML-33/1* | *2.69* | *685* | *27* | | | | |
| HZM-1.1 | 2.90 | 796 | 32 | 685 | 40 | 644 | 41 |
| HZM-1.2 | 2.91 | 802 | 32 | 795 | 40 | 708 | 29 |
| HZM-1.3 | 2.93 | 810 | 32 | 975 | 90 | 869 | 94 |
| *ML-33* | *2.94* | *819* | *33* | | | | |
| *ML-32* | *3.29* | *985* | *39* | | | | |
| *ML-31/1* | *3.54* | *1248* | *50* | | | | |
| HZM-2.2+3 | 4.01 | 1569 | 63 | 1565 | 55 | 1451 | 57 |
| *ML-31* | *4.17* | *1710* | *68* | | | | |
| *ML-30* | *4.29* | *1789* | *72* | | | | |
| *ML-29* | *4.59* | *1984* | *79* | | | | |
| *ML-28* | *4.91* | *2219* | *89* | | | | |
| HZM-3.1 | 5.16 | 2433 | 97 | 2405 | 60 | 2469 | 112 |
| *ML-27* | *5.17* | *2449* | *98* | | | | |
| HZM-3.3* | 5.19 | 2450 | 98 | 2750 | 60 | (2850) | 66 |
| *ML-26* | *5.43* | *2593* | *104* | | | | |
| HZM-23* | 5.45 | 2595 | 104 | 2720 | 60 | (2826) | 58 |
| HZM-24 | 5.61 | 2754 | 110 | 2620 | 65 | 2743 | 101 |
| *ML-25/1* | *5.77* | *3147* | *121* | | | | |
| HZM-25 | 5.97 | 3704 | 136 | 3465 | 70 | 3730 | 96 |
| *ML-25* | *6.11* | *3992* | *143* | | | | |
| *ML-24* | *6.21* | *4420* | *154* | | | | |
| HZM-26* | 6.23 | 4616 | 159 | 4100 | 90 | 4624 | 127 |
| *ML-23* | *6.51* | *5083* | *171* | | | | |
| *ML-22* | *6.68* | *5286* | *177* | | | | |
| HZM-4.1 | 6.69 | 5334 | 177 | 4575 | 65 | 5243 | 131 |
| HZM-4.2 | 6.70 | 5359 | 177 | 4730 | 70 | 5462 | 85 |
| HZM-4.3 | 6.71 | 5385 | 178 | 4675 | 70 | 5409 | 95 |
| *ML-21* | *6.78* | *5520* | *179* | | | | |
| *ML-20* | *6.84* | *5619* | *179* | | | | |
| *ML-19* | *7.05* | *5977* | *182* | | | | |
| *ML-18/2* | *7.51* | *6328* | *184* | | | | |
| *ML-18/1* | *7.70* | *6590* | *186* | | | | |
| *ML-18* | *8.06* | *7274* | *191* | | | | |
| HZM-5.3 | 8.13 | 7428 | 192 | 6455 | 70 | 7363 | 68 |
| *ML-17/3* | *8.42* | *7870* | *195* | | | | |
| *ML-17/2* | *8.72* | *8338* | *198* | | | | |





| | | | | | | | |
|---|---|---|---|---|---|---|---|
| ML-17/1 | 9.03 | 8943 | 203 | | | | |
| HZM-6.1 | 9.33 | 9649 | 207 | 8800 | 95 | 9851 | 170 |
| ML-17 | 9.40 | 9746 | 208 | | | | |
| ML-16 | 9.66 | 10169 | 211 | | | | |
| HZM-7 | 9.92 | 10464** | 213 | 9465 | 45 | 10705 | 130 |
| ML-15 | 9.92 | 10554 | 214 | | | | |
| ML-14 | 10.03 | 10681 | 215 | | | | |
| HZM-8 | 10.07 | 10708 | 215 | 9495 | 55 | 10773 | 148 |
| ML-13 | 10.24 | 10999 | 217 | | | | |
| HZM-9 (UMT) | 10.25 | 11008 | 217 | 9560 | 49 | 10923 | 121 |
| HZM-40 | 10.27 | 11048 | 217 | 9550 | 80 | 10901 | 148 |
| HZM-41 | 10.33 | 11109 | 218 | 9665 | 100 | 10998 | 154 |
| HZM-42 | 10.38 | 11145 | 218 | 9565 | 100 | 10912 | 160 |
| HZM-43 | 10.46 | 11226 | 219 | 9830 | 100 | 11264 | 178 |
| ML-12 | 10.48 | 11232 | 219 | | | | |
| HZM-44 | 10.52 | 11267 | 219 | 9805 | 190 | 11243 | 329 |
| HZM-45 | 10.59 | 11322 | 219 | 9905 | 80 | 11357 | 138 |
| HZM-46 | 10.64 | 11357** | 219 | 10060 | 80 | 11584 | 159 |
| HZM-10.1 | 10.67 | 11339** | 219 | 10085 | 80 | 11630 | 165 |
| HZM-47 | 10.70 | 11400 | 220 | 10110 | 110 | 11680 | 231 |
| ML-11 | 10.73 | 11453 | 220 | | | | |
| HZM-48 | 10.78 | 11534 | 221 | 11040 | 140 | (12959) | 120 |
| HZM-50 | 10.99 | 11942 | 241 | 10080 | 110 | 11628 | 214 |
| ML-9 | 11.02 | 11943 | 241 | | | | |
| HZM-12 | 11.10 | 12354 | 266 | 10520 | 90 | 12509 | 181 |
| HZM-51 | 11.14 | 12530 | 276 | 10350 | 90 | 12203 | 194 |
| ML-8 | 11.20 | 12578 | 279 | | | | |
| HZM-13* | 11.38 | 12769 | 290 | 11295 | 85 | (13197) | 74 |
| ML-7 | 11.41 | 12778 | 291 | | | | |
| HZM-14* | 11.48 | 12861 | 296 | 11780 | 100 | (13647) | 112 |
| ML-6 | 11.56 | 12880 | 297 | | | | |
| ML-5 | 11.70 | 12880 | 297 | | | | |
| HZM-30 | 11.74 | 12925 | 299 | 11250 | 110 | 13158 | 109 |
| ML-4 | 11.86 | 13087 | 309 | | | | |
| HZM-16* | 12.03 | 13130 | 311 | 13140 | 140 | (15766) | 212 |
| HZM-32 | 12.19 | 13445 | 330 | 11770 | 135 | 13642 | 150 |
| HZM-17 | 12.26 | 13472 | 332 | 12100 | 110 | 13984 | 183 |
| ML-3 | 12.40 | 13646** | 339 | | | | |
| HZM-35 | 12.78 | 13985 | 362 | 12570 | 130 | 14858 | 286 |
| ML-2 | 12.86 | 14152** | 369 | | | | |
| HZM-18 | 12.90 | 14156 | 372 | 12430 | 110 | 14586 | 249 |
| ML-1 | 12.90 | 14156** | 372 | | | | |
| HZM-100*** | 12.92 | 14157 | 372 | 12380 | 85 | 14492 | 228 |
| HZM-19 | 12.93 | 14158 | 372 | 12555 | 80 | 14879 | 221 |


*Dates described to contain reworked organic material or being fractionated during graphitization (see Hajdas et al.,*
*1995).*
*** VT-99 dates excluded from modelling due to inconsistencies in documentation.*
**** unpublished radiocarbon age (KIA-1460)*





*Table A3: Core section depths of the composite profile HZM19 with resulting composite end depths for each core.*

| Core section | From [mm] | To [mm] | Length [mm] | Composite core section end depth [mm] |
|---|---|---|---|---|
| HZM19_07_01 | 138 | 800 | 662 | 662 |
| HZM19_08_01 | 305 | 755 | 451 | 1113 |
| HZM19_07_02 | 243 | 924 | 681 | 1794 |
| HZM19_08_02 | 380 | 839 | 459 | 2254 |
| HZM19_07_03 | 229 | 912 | 683 | 2936 |
| HZM19_08_03 | 375 | 714 | 339 | 3275 |
| HZM19_07_04 | 243 | 800 | 557 | 3833 |
| HZM19_08_04 | 235 | 994 | 759 | 4592 |
| HZM19_10_01 | 90 | 913 | 823 | 5415 |
| HZM19_08_05 | 630 | 930 | 299 | 5715 |
| HZM19_10_02 | 183 | 877 | 693 | 6409 |
| HZM19_08_06 | 596 | 957 | 361 | 6770 |
| HZM19_10_03 | 87 | 827 | 740 | 7510 |
| HZM19_08_07 | 562 | 971 | 409 | 7919 |
| HZM19_10_04 | 179 | 870 | 691 | 8611 |
| HZM19_08_08 | 641 | 967 | 326 | 8937 |
| HZM19_10_05 | 137 | 859 | 722 | 9659 |
| HZM19_11_06 | 395 | 655 | 260 | 9919 |
| HZM19_08_10 | 35 | 974 | 939 | 10859 |
| Technical gap | | | 129 | 10988 |
| HZM19_10_07 | 30 | 810 | 780 | 11768 |
| HZM19_11_07 | 710 | 1012 | 302 | 12071 |
| HZM19_10_08 | 72 | 902 | 830 | 12902 |
| HZM19_11_08 | 326 | 1245 | 919 | 13822 |
| HZM19_07_17 | 100 | 920 | 820 | 14643 |








*Table A4: Core section and composite depths of lithozones H1 to H12 for HZM19*

| Lithozone | From | | | To | | | Biozone | Human phase |
|---|---|---|---|---|---|---|---|---|
| | Section | Section depth [mm] | Composite depth [mm] | Section | Section depth [mm] | Composite depth [mm] | | |
| H12 | HZM19_07_01 | 138 | 11 | HZM19_08_01 | 700 | 1057 | Subatlantic | Last century |
| H11 | HZM19_08_01 | 700 | 1058 | HZM19_08_03 | 520 | 3081 | Subatlantic | Middle Ages / Little Ice Age |
| H10 | HZM19_08_03 | 520 | 3081 | HZM19_08_04 | 710 | 4308 | Subatlantic | Migration Period / Early Middle Ages |
| H9 | HZM19_08_04 | 710 | 4308 | HZM19_08_05 | 750 | 5535 | Subatlantic | Iron Age / Roman Period |
| H8 | HZM19_08_05 | 750 | 5535 | HZM19_10_05 | 480 | 9280 | Subboreal/Atlantic | |
| H7 | HZM19_10_05 | 480 | 9280 | HZM19_11_06 | 588 | 9852 | Boreal | |
| H6 | HZM19_11_06 | 588 | 9852 | HZM19_08_10 | 140 | 10025 | Preboreal | |
| H5 | HZM19_08_10 | 140 | 10025 | HZM19_08_10 | 860 | 10745 | Preboreal | |
| H4 | HZM19_08_10 | 860 | 10745 | HZM19_08_10 | 970 | 10855 | Preboreal | |
| H3 | HZM19_08_10 | 970 | 10855 | HZM19_10_07 | 300 | 11258 | Younger Dryas | |
| H2 | HZM19_10_07 | 300 | 11258 | HZM19_10_08 | 859 | 12859 | Bölling/Alleröd | |
| H1 | HZM19_10_08 | 860 | 12860 | HZM19_07_17 | 920 | 14643 | Pleniclacial (Late Weichselian) | |



*Table A5: Additional dates for the HZM19 chronology with composite depths, ages (cal. BP) and errors used for Bacon*
*calculations. LST age with error is from Reinig et al. (2020).*

| Event | HZM19 comp. depth (cm) | Age (cal. BP) | error |
|---|---|---|---|
| Sediment surface | 0.00 | -69 | 1 |
| Chernobyl accident | 47.20* | -36 | 1 |
| Maximum atomic bomb tests | 67.20* | -13 | 1 |
| First atomic bomb tests | 79.20* | 0 | 1 |
| Laacher See Tephra | 1160.00 | 13,006 | 9 |

*\* 22 cm subtracted due to slump event documented by Pb-210 data.*



*Table A6: Age estimations for VT-99 and Models A-D with their 95% confidence intervals in brackets for Ulmener Maar*
*Tephra (UMT), Younger Dryas/Preboreal-transition (YD/PB), YD duration, Allerød/Younger Dryas-transition (AL/YD),*
*predicted YD hiatus with duration and position, Laacher See Tephra (LST), Maximum model age at 12.93 m with its mean*
*and maximum age ranges and position of the maximum age range and maximum difference between VT-99 and each of*
*the model ranges.*

| Chronology | VT-99 | A | B | C | D |
|---|---|---|---|---|---|
| Age of UMT | 10999 [10782, 11216] | 10961 [10784, 11090] | 10965 [10787, 11093] | 10952 [10788, 11067] | 10981 [10829, 11088] |
| YD/PB transition | 11632 | 11674 [11461, 11965] | 11723 [11486, 12070] | 11682 [11494, 11913] | 11655 [11499, 11845] |
| YD duration | 974 | 1038 | 1014 | 1012 | 1073 |
| AL/YD transition | 12606 | 12712 [12517, 12880] | 12737 [12562, 12880] | 12694 [12475, 12869] | 12728 [12595, 12838] |
| Duration of YD hiatus | 320 | 623 | 603 | 583 | 686 |
| End of YD hiatus | 12025 | 11863 [11571, 12269] | 11952 [11623, 12502] | 11901 [11646, 12207] | 11854 [11651, 12098] |
| Age of LST | 12880 [12583, 13177] | 13010 [12984, 13042] | 13010 [12985, 13043] | 13009 [12984, 13037] | 13011 [12984, 13043] |
| Maximum model age (at 12.93 m) | 14158 [13786, 14530] | 14615 [14339, 14926] | 14456 [14236, 14749] | 14614 [14332, 14919] | 14485 [14287, 14721] |
| Mean age range | 282 | 468 | 456 | 329 | 229 |
| Maximum age range | 744 | 1056 | 1064 | 749 | 447 |
| Max. age range position (m) | 12.93 | 8.86 | 8.78 | 9.18 | 11.09 |
| Maximum difference to VT-99 age range | 0 | 655 | 665 | 401 | 0 |





## Data Availability

The results of the different age-depth models carried out for the lacustrine sediment record
from Holzmaar are accessible via the PANGAEA data archiving and publication system at
https://doi.org/10.1594/PANGAEA.xxxxxx.

## Author contributions

SB and BZ conducted the fieldwork and conceptualized the study. SB described and sampled the
sediment, modified and run the Bayesian age-depth models, visualized the data and drafted the
first version of the manuscript. WT measured and interpreted lead and cesium data. All authors
contributed to the writing and to revising of the manuscript.

## Competing interests

The contact author declares that neither she nor her co-authors have any competing interests.

## Disclaimer

## Acknowledgments

We like to thank Christian Ohlendorf, Rafael Stiens and An-Sheng Lee for participating in the
coring campaign of 2019 and also for subsequent help with core opening, sediment preparations
and scanning in the GEOPOLAR lab. Furthermore, we want to thank Maarten Blaauw, Arne
Ramisch and Alicja Bonk for helpful discussions.

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
