# Peer review of "Bayesian age-depth modelling applied to varve and radiometric dating to optimize the transfer of an existing high-resolution chronology to a new composite sediment profile from Holzmaar"

_Geochronology, 2022_

## Author Comment (AC3)

**Dear Referee #1, we thank you a lot for very constructive suggestions. In the following we response (in bold) to all of your detailed comments (in italics).**

*Although this is a good methodological exercise, the objectives, methodological implications of this study are not clear to me. I believe the main purpose when applying Bayesian statistics to perform an age-depth model is to combine as much chronological and stratigraphic information as possible to get the best dating and probabilistic estimates of age uncertainties. According to that, the conclusions of the study i) all the Bayesian models improve the accuracy and precision of previous age estimates and ii) Method D (the one which includes more detailed chronological and stratigraphic information) is the best approach, were fairly predictable results. On the other hand, if the main objective is to transfer an existing chronology to a new composite sediment profile as indicated in the title, the high-resolution stratigraphic correlation using marker layers should be enough, especially in varved sediments.*

**Our study compares different methods to integrate radiocarbon dates and varve-dated marker layers using a Bayesian approach with the Bacon package. However, we did not conclude that all of the presented models improve accuracy and precision of the previously published varve chronology VT-99 (i), especially not in the youngest part and in parts with less radiocarbon ages (as shown by Model D, section 3). We also disagree, that the good performance of Model D was predictable (ii), as Model B, C and D contain varve information and radiocarbon ages merged in different ways.**

*Holzmaar is one of the best studied varved records in the world and ongoing studies of these sediments (e.g. Garcia et al., 2022) are an example of the use of new methodological approaches in palaeolimnology with potential for significant impact on regional palaeoenvironmental and palaeoclimate research. I strongly support the publication of a new improved and robust chronology for this site but, in my opinion, the manuscript needs some changes in the structure and content before publication. Below are my suggestions:*

*(1) I really think the manuscript would benefit from a better description of the objectives and I propose two options to do so that, I hope, can help:*

*Option 1 (the most sensible to me). The main goal is to improve the existing Holzmaar chronology and to transfer it to the new composite profile. In this case, I would focus on a better description of the correlation between the old and new composite profiles. I miss a figure showing the two composite profiles with the position of the marker layers, radiocarbon dates, tephra layer and prior information used in Method D. I would focus on the comparison of the VT-99 chronology, the radiocarbon chronology (Method A) and the integrated Bayesian chronology (Method D) and I would discuss the new chronology (Method D) reporting age uncertainties and new age estimates for the LST, UMT and climatic transitions based on Method D as described in Section 3.2.4. This information might be relevant for other researchers working on this region.*

*Option 2. The main goal is to discuss the best approach for Bayesian age-depth modelling in varved sediments in general and Holzmaar in particular. This option implies additional work. I would suggest a comparison of Method D in Bacon with a Bayesian age-depth model in Oxcal using the same chronological information. This would allow discussing the pros and cons of the two approaches for varved sediments, which would be a significant contribution to the community.*

Thank you for these two options, which we appreciate. We agree with the need for a figure showing the composite profiles from before and after applying the Bayesian modelling. The aim of our study was to compare different varve and radiocarbon integration methods, to find the best way of adapting and updating an existing varve chronology to a new profile. We are aware of the highly specialized topic of this research, but we strongly believe that in the future more projects will face situations like this. Therefore, we do not agree excluding any of the tested approaches here, because only this approach allows future researcher to compare their data directly instead of testing them for their own and once again. We completely agree with OxCal being an additional option. In our case, we want to give an overview of the different approaches using Bacon. A comparison of OxCal with Bacon was in our minds. However, implementing this into our study would increase the size of the manuscript distinctly.

Therefore, we will implement a new figure comparing HZM-B/C with HZM19 that shows the marker layer positions, depths of radiocarbon ages, tephras and other chronological markers used in our study. However, we prefer not to change our general approach. Additional arguments are presented below.

*(2) The structure of the manuscript needs improvements as follows:*

*Introduction: the introduction does not provide sufficient background information to understand the issue addressed and the significance of this study. I found the missed information in other parts of the manuscript though, so I think this is just to move some paragraph into this section.*

We appreciate your thoughtful suggestions and address each individual point as followed:

- *Paragraph 1, 2 and 3 (line 37-61) need to be supported by references.*

We agree with your suggestion and will add more references (highlighted in yellow) to lines 37-61:

"Terrestrial archives from lakes have the potential to provide information about climate and the human history of its catchment area beyond instrumental and historical data (Berglund, 1986; Last and Smol, 2001a, b; Cohen, 2003). In the late 1980s, gravity coring (Kelts et al., 1986) piston coring (Nesje et al., 1987; Wright et al., 1984) and freeze coring techniques (Renberg and Hansson, 1993) for lacustrine sediment records have improved tremendously allowing a better quality of sediments to be recovered from modern lakes. Since then, the new fields of limnogeology and paleolimnology flourished with increasing demand of societies for documentation of natural background data related to questions around acid rain (e.g. Battarbee et al., 1990), environmental pollution (e.g. Renberg et al., 1994) and more and more with a focus on global climate change (e.g. Jenny et al., 2019).

To provide such information not only on local scales but also on larger regional to global scales, investigations from different sites need to be compared and linked. However, such correlations are only successful if the contributing archives are based on robust chronologies. Therefore, precise and reliable age-depth models are the basis for sedimentary investigations and reconstructions of environmental and climatic changes of the past, as they ensure intra-site comparability and enable recognition of larger scale patterns. A reliable chronology should be based on a combination of different dating techniques (multiple dating approach) such as radiometric dating, well-known events such as tephra layers (Turkey and Lowe, 2001), historic data (e.g., flood events) or varve counting. The term "varve" (Swedish: layer) was first introduced by De Geer (1912) for outcrops with proglacial sediments and describes finely laminated sediment structures with annual origin. The alternating pale and dark layers are driven by seasonal changes in temperature and precipitation that cause different chemical and biological processes within the lake and its catchment area. When anoxic conditions at the sediment-water-interface are given at least seasonally, i.e. no bioturbation destroys laminations, varves are preserved and provide high-resolution and precise chronologies in calendar years (Zolitschka et al., 2015; Lamoureux, 2001).

Until the 1980s, varve chronologies were the only option for calendar-year chronologies for sediment records, while AMS radiocarbon dating was still in its infancy and calibration of radiocarbon ages was restricted to tree rings of the Middle and Late Holocene, if at all applied (Pearson et al., 1977; Olsson, 1986)."

- *Sub-subsection 2.3.4 "Bayesian age-depth modelling" (in varved sediments?) should be added to the Introduction (line 65). And after that, I would add the sentence in line 94-99. I would provide more details about the main reasons to choose Bacon based on the information about Bacon and Oxcal you give in sub-subsection 2.3.4.*

We agree of rearranging these chapters, but will implement it in line 71 instead of 65. We include the first part (lines 239-264) of chapter 2.3.4, as the remaining part contains too detailed information for an introduction. We attached lines 94-99 to the end of it and added more details about our decision as followed:

"In this study, we focus on varve-counting integration methods using Bacon (rBacon version 2.5.7; Blaauw et al., 2021; Blaauw and Christen, 2011) for the R programming language (version 4.1.1; R Core Team, 2021), as it is one of the most often used software package in paleo studies and provides many different ways for implementing information."

After lines 94-99 we continue:

**"As Bacon provides many different options to incorporate information into the age-depth model, in the literature only few approaches are provided integrating varve and radiocarbon ages (Bonk et al., 2021; Vandergoes et al., 2018; Shanahan et al., 2012). For that reason, we summarize these approaches and compare them directly with each other. This will lead to faster decisions for future studies facing a comparable situation."**

- *Information provided in line 70 -82 is duplicated in Section 2.3.1. I suggest to removed it from the introduction.*

**We agree and removed it from the introduction.**

- *Aims and Objectives are not clear (see comment 1 above)*

**Thank you for this clarification. We added our aims and objectives by formulating the last part of the introduction as follows (line 100-107):**

**"The aim of our study is to transfer and optimize the existing varve chronology from HZM-B/C to the new sediment record HZM19. In addition, we offer an overview about different approaches for age-depth modelling and their effects on model outcomes to researchers who face comparable challenges, thus supporting their decision making.**

**For this reason, we discuss the possibilities of integrating and improving the chronology by combining the varve chronology with modelling approaches using Bacon. This is accomplished by testing and comparing integration methods with regard to accuracy and precision obtained from the interpolated varve chronology itself and from a Bayesian model without any varve information relying on radiocarbon dates only.**

**With this integration of all age information we produce the most reliable age estimations for the HZM19 record: VT-22. Based on this best model outcome, this master chronology of VT-22 serves as the chronological backbone for ongoing and future biological, geochemical and geophysical investigations conducted with the new Holzmaar sediment cores (e.g. García et al., 2022).**

*Material and Methods:*

- *Subsection 2.1 "Regional Settings" should be under an independent section. I suggest a new Section 2 on "Regional settings and the Holzmaar sediment record". which includes (1) the current subsection 2.1 "Regional settings", (2) subsection 2.2. "Holzmaar lithology" where you provide information about the published lithology from old cores (Zolitscka 1998 a and b) as described in subsection 3.1. And (3) Subsection 2.3 "Previous Holzmaar chronology" which corresponds to the current su-subsection 2.3.1. Material and Methods would be Section 3 then.*

**We completely agree with the rearrangement of these chapters and adapted them as suggested. The new chapter structure will be as followed: … -**

**2. Regional setting and the Holzmaar sediment record**
**2.1 Regional setting**
**2.2 Holzmaar lithology**
**2.3 Previous Holzmaar chronology**
**3. Materials and Methods**
**3.1 Sediment core collection**
**3.2 Chronology**
**3.2.1 Pb-210 and Cs-137 dating**
**3.2.2 Bayesian age-depth modelling**
**4. Results and Interpretation**
**4.1 Transfer of VT-99 to HZM19**
**4.2 Pb-210 and Cs-137 dating**
**4.3 Age-depth modelling**
**4.4 Comparison of model output with VT-99**
**4.5 Comparison of model output with the common isochrones**
**… (as before).**

- *Line 152: please provide information of the length of the cores, how many parallel cores you collected, distance between them and the sediment depths they cover.*

**We provided the requested information as follows:**

**"The coring locations are distributed evenly along a 12 m-long transect with 4 to 4.4 m distance between coring locations. The recovered sediment cores have lengths of 2 m (HZM19-07, -08, -10) and 3 m (HZM19-11), which have been split in the field into 1 and 1.5 m-long sections, respectively. In total, HZM19-07 covers a sediment depth of 15.5 m (0-15.5 m), while the other sites provided different depth ranges: HZM19-08 (0.25 – 10 m), HZM19-10 (4 – 14 m) and HZM19-11 (1 – 19 m)."**

- *Line 157: please say how many marker layers you have used for correlation*

**For correlation of the cores we used 48 distinct correlation marker layers. We will include this number into line 157 and include a table with related sections depths into the appendix.**

- *I would say that Sub-subsection 2.3.2 "Transfer of VT-99 to HZM19" should be part of the results.*

**We agree and added subsection 2.3.2 to the result section (see above).**

- *Line 265-266. Reference is needed.*

**We added four different examples from the literature providing methods of varve and radiocarbon integration with Bayesian modelling approaches: Bonk et al., 2021; Vandergoes et al., 2018; Shanahan et al., 2012; Fortin et al., 2019**

*Results and Interpretation:*

- *It makes more sense to me that the lithozones are described as previous work (see my comment above re a new Section 2). Subsection 3.1 should focus on the correlation of the HZM 99 and HZM 19 composite profile and the transfer of the varve chronology (current subsection 2.3.2). It would be good to see in a figure the two composite profile, the stratigraphic position of the marker layers, radiocarbon dates, hiatus, etc and both the VT-99 varve age-depth profile and a 14C chronology.*

**We completely agree with this suggestion. With the new structure (see above), the Results and Interpretation will become chapter 4. Thus, chapter 4.1 will focus on the transfer and correlation of VT-99 from the old cores as recommended. To achieve this, we combined previous chapters 2.3.2 "Transfer of VT-99 to HZM19" and 3.2.2 "Varve time and independent chronology". We will also add a figure to supplementary material showing the old composite sediment profile HZM-B/C and the new HZM19 together with positions of marker layers:**

[Figure]

*Figure A1: Correlation of HZM84-B/C and HZM19. Positions of marker layers (ML indicated to the left) are marked as solid lines and connected by dotted lines between both profiles. Positions of radiocarbon dates (numbers indicated in rectangular*

*boxes to the right) are marked as solid circles. Grey dotted horizontal lines refer to Cs-137 dated depths. Positions of Ulmener Maar Tephra (UMT), Laacher See Tephra (LST) and the technical gap are indicated.*

- *I would call subsection 3.2 "New chronological information" and make sub-subsection 3.2.3 and new subsection 3.3 "Age-depth modelling".*

**Due to the rearrangement of chapters mentioned above, chapter 4.2 will "4. 2 Pb-210 and Cs-137 dating".**

- *Sub-subsection 3.2.4 and sub-subsection 3.2.5 should be subsection 3.5 and 3.4, respectively. Foucssing on Method D only (I would delete Mehod B and C from the manuscript), first describe the improvements in dating and age uncertainty using the best Bayesian model (Method D) with respect to the varve chronology (VT-99) and radiocarbon chronology (Method A) (using the text in sub-subsection 3.2.5). Second, report new age estimates for the tephra layers. As these tephra layers, especially the LST, have been used for synchronising records and the estimation of the duration on the YD in different European sites (e.g Wulf et al., 2013), a revised age estimate with a reduced age uncertainty from HZM might be very useful.*

**We rearranged the chapters to: 4.4 Comparison of the model output with VT-99 and 4.5 Comparison of model output with the common isochrones. We do not exclude Model B and C for reasons mentions above. We agree with the statement that tephra layers are very important chronological marker layers for several studies in related fields. However, we incorporated the latest LST age estimation into the calculation for all models. Thus, our date is very close to the published age by Reinig et al., 2021. As we used this age for our modelling, the new LST age is not independent. We reported our outcome for both tephra layers but will focus on the age difference between both isochrones. This was also recommended by Reviewer #2.**

**References**

[revised manuscript text omitted]

---

## Author Comment (AC4)

**Dear Referee #2, we thank you a lot for very constructive suggestions. In the following we response (in bold) to all of your comments (in italics).**

*The manuscript presents a considerable effort the authors put into transferring the previously obtained chronology for Lake Holzmaar to newly recovered sediment cores. The chronology for Lake Holzmaar is a unique one, with high resolution, based on varve counting, radiocarbon and isotope measurements. The authors carefully evaluate the reliability and accuracy of all the results, and this is one of the strongest points of this work.*

*The progress presented by this manuscript concerns testing four different approaches to build chronology for HZM19 record in a quantitative way – concerning the precision, accuracy, and comparison with other records for distinct events, like tephra layers and biozone boundaries. Hardly ever this kind of approach is published, and typically only one, "the best" or "the chosen" age-depth model is presented in publications. Usually there is no space to discuss the reasons behind the choice and address questions of age-depth model methodology in papers focusing on proxy-interpretation. As such, I think "Geochronology" is the right journal to publish this kind of study. This manuscript can also be regarded as a guide to future research teams which may face similar challenge in the future.*

**Many thanks for these motivating words. We appreciate your opinion very much and agree that methodological studies are quite rare and up to know no best approach to follow has been published. We are convinced that this will change in upcoming years and think that our study will contribute to this development.**

*The authors preformed the modelling with use of Bacon code - the modern, but well-established tool for Bayesian age-depth modelling. They proved an excellent knowledge and know-how about using the prior information in a process of age-depth modelling, which I know from my experience is not a trivial task. On the other hand, "playing" with priors may sometimes be used in an inappropriate way, e.g. to get the modelled age matching some expectations or get unrealistic precision, but here the authors convinced me they set the parameters to realistic and justified values.*

*The exhaustive Introduction provides a valuable and complete context of Lake Holzmaar chronology challenges and improvements. Discussion of the results is well-balanced, and based on scientific evidence, also taking into consideration the previously obtained data, with appropriate references. Some minor issues I address in "Specific comments".*

**Thank you for this nice feedback. We agree that the settings using Bacon can be too difficult or incomprehensible for beginners in this field. However, they are very useful as soon as comprehended to a certain degree. Studies like ours might also help Bacon beginners to understand the different effects of parameter settings and what might be best for their own case study.**

*The manuscript follows the classical structure (introduction-methods-results-discussion-conclusion), which is appropriate and clear. Some of the figures and all the tables are presented in Appendix, which is fine, although the Fig. A3 is cited 16 times (!) in the manuscript text, and I suggest moving it to the core of the paper. The quality of figures and tables is good, I have some minor remarks – see technical part of the review.*

*In my non-native-speaker opinion the language reads fluently.*

**Yes, we completely agree that Figure A3 should be shifted into the manuscript. We will implement it as Figure 5 into the manuscript.**

*Pages 13-14*

*The ages derived from of 137Cs peaks are clear, I have no doubts about it, but why the slump, clearly present in 210Pb and lithology, is not demonstrated in 137Cs data? If I imagine cutting the slump section out of the 137Cs profile, it would't look as nicely monotonous as it is now. Do authors have any thoughts on that?*

**We apologize for making this point not clear enough. Here is our answer:**

**The question why the slump is not clearly demonstrated in Cs-137 activity is difficult to answer. At this section, very slow decrease in Cs-137 activity towards the top is observed. After removing the slump from the profile, there will be a slight shift to lower values but similar variations are observed below the slump section as well (between 60 and 35 cm sediment depth, see Fig. 3). It is impossible to indicate a direct reason for these shifts because small-scale incidental slumps caused by artificial disturbance may produce random variability in the Cs-137 profile.**

**Generally, the Cs-137 profile is smooth lacking sharp peaks, which very precisely indicates two chronostratigraphic markers, i.e. 1963 and 1986. However, there are blurry peaks, which can be interpreted as chronostratigraphic information. The reasons for this blurring may be high sedimentation rates at the coring location. Additionally, there is substantial contribution of horizontal replacements of sediments in the surface section due to redeposition caused by drifting of the monitoring buoy for meteorological data anchored in the central part of the lake. This buoy was installed in 1994 and removed from the lake in 2016. We know that several times the buoy drifted almost to the lake shore due to very strong winds. Displacement of the heavy anchor must have caused sediment resuspension and disturbances of surface sediments at the lake bottom.**

*Page 15*

*In line 377 authors state they excluded two 14C results (HZM-46 and HZM-10.1) from a list of marker layers, due to "inconsistencies in documentation". As such I would expect they are not included in any discussion and conclusion, but then in line 391 HZM-46 is referenced to – I suggest to leave the depth info only in line 391.*

**We agree with this suggestion and changed the text accordingly referring only to the depth value in line 391.**

*Page 18 Line 465-466.*

*I wonder about the reasons for a high noise in Model C, do the authors have some explanation for this observation? My guess would be lot of data with high density per core length, and relatively small uncertainties.*

**Thank you for this question. The instability of the MCMC iterations in Model C must have to do with the implementation of marker layers with normally distributed ages. We assumed that the implementation would have a positive effect on the stability of the model, but we observed the opposite. The approach seems to increase the accuracy of the non-radiocarbon dated depths with unequal jumps from these depths to the radiocarbon-dated depths, which are not directly visible in the output plot. Very high noise in iterations is normally observed, when the model calculates very different ages for each iteration. The differences have to be smaller scaled. We cannot add further explanations here and agree with your idea of high density per core length and small uncertainties of the input data.**

*Page 19, lines 495-499*

*Anchoring of the Section 3 was first mentioned in lines 289-290, here the explanation is provided plus reference to Fig. A4. Honestly, I was not able to understand the reason and way to sum the probabilities for four completely different radiocarbon results. How the ages of HZM5.3, HZM6.1 and HZM7 were shifted to form the PDFs presented in Fig. A4A? Please clarify this part of calculations. Was the age of HZM4.3 not sufficient to anchor the Section 3?*

**We agree with the need for a better explanation here and apologize for unclear formulation. For model D we basically followed the approach by Bonk et al., 2021. When we tried to transfer their method to our study, we struggled at the same part and had to contact the authors to clarify their approach of connecting the varve chronology part to the Bayesian model part. After further explanation by them, we were able to apply their approach. First, we provide more details to the calculation, then explain why we prefer this approach and finally how we implement a better explanation into the manuscript.**

**What can be seen in Figure A4A are not shifted radiocarbon ages. The x-axis refers to each tested anchor for the varve chronology in section 3 (within the age range of HZM-4.3), while the y-axis documents which probability level of each single radiocarbon age is matched if the varve chronology is anchored at age x. To find the best position along x, we simply summed all the different matched probabilities for each radiocarbon age to obtain the sum for each anchor shift along x (black line). To summarize, Figure A4A only shows, which probabilities of the calibrated radiocarbon ages are matched when we shift the anchor along the HZM-4.3 age range.**

**We are sure that this approach leads to the best result, as it updates the varve chronology in section 3, while considering the latest calibration curve. Furthermore, the calibrated median age of HZM-4.3 (5409 +- 95 cal. BP) is slightly younger than the basal age of the model calculated in section 2 (5419 +- 165 cal. BP), whereas they agree within their age ranges. The same issue occurs with the original VT-99 age of HZM-4.3 (5389 +- 178). By accepting the calculated anchor of 5450 cal. BP, we increase the gap between section 2 and 3, but can provide an age model without inverse age-depth relationships and at the same time decrease the gap between sections 3 and 4 (10.578 to 10.663 cal. BP)**

**To make this point clearer for the readership of the manuscript, we implemented following changes: We changed:**

1) **the x-axis title of Figure A4A to "Tested anchor age (Age cal yr BP)".**

2) **the figure caption to: "Figure A4: Calculations for the floating VT-99 chronology of Model D, section 3. A: Calculation of the anchoring age for the varve chronology based on matched and summed calibrated probability density function values of all radiocarbon samples within this section. The maximum summed probability occurs at an anchor age of 5450 cal BP. B: Original VT-99 (black) vs. floating VT-99 (+65 years, red dotted line?) with calibrated radiocarbon samples vs. depth.**

*Page 21*

*If the age of LST is implemented as a marker (as stated in line 530) then it should not be derived from the model (as in line 531). I suggest deleting "and LST" in line 531 or rephrasing this sentence, and still the following paragraph discussing the interval between UMT and LST is valid.*

*Similar conclusion is provided on page 24, line 615 – please avoid circular reasoning*

**We agree and adapted your suggestion to delete "and LST" from line 531 and 615.**

*Page 22*

*I have a feeling the whole presentation on YD boundaries and duration, and comparison with other records, would benefit from some graphical illustration in addition to numbers cited in text and given in Table A6. Please consider adding such plot.*

**We agree and implemented the following figure into the manuscript:**

[Figure]

*Figure A5: Close-up plots for the Lateglacial / Early Holocene transition for Model A, B, C and D with VT-99 mean age (black solid line) and error (shaded in gray) for comparison. Horizontal lines as labelled in (a). Vertical lines refer to the Younger Dryas transitions for each Model (solid lines), while dotted lines refer to mean ages derived by different sites (Lake Gosciaz in blue: Bonk et al., 2021; Meerfelder Maar in red: Brauer et al., 1999).*

*Technical corrections*

*Line 10*

*Abstract, first line "This study gives an overview of different varve integration methods with Bacon." sounds colloquial, I suggest elaborating, consider e.g. "…different methods to integrate information from varve chronology, radiometric measurements in Bayesian tool Bacon…"*

**We accept your suggestion and integrated your formulation in the text.**

*Line 79 and elswhere*

*Please correct the referenced name to "Bronk Ramsey, 2009" as this is a correct two-part surname for Christopher Bronk Ramsey*

**We apologize this error and corrected this mistake throughout the whole manuscript.**

*Line 149-153*

*Please include a brief information about the total length of the recovered cores, maybe refer to Fig. 2?*

**Thank you for your suggestion. This was also suggested by reviewer #1 and we have added this information as follows:**

**"The coring locations are distributed evenly along a 12 m-long transect with 4 to 4.4 m distance between coring locations. The recovered sediment cores have lengths of 2 m (HZM19-07, -08, -10) and 3 m (HZM19-11), which have been split in the field into 1 and 1.5 m-long sections, respectively. In total, HZM19-07 covers a sediment depth of 15.5 m (0-15.5 m), while the other sites provided different depth ranges: HZM19-08 (0.25 – 10 m), HZM19-10 (4 – 14 m) and HZM19-11 (1 – 19 m).**

*Line 181*

*Change "Spectroscopy" to "Spectrometry", the correct name for the AMS technique*

**We changed this as suggested in the revised manuscript.**

*Line 317*

*If possible, please enlarge Fig. 2 to full-page scale, would be easier to read*

**We enlarged Figure 2 to full page scale.**

*Line 422*

*Change "+ -" to "±"*

**We corrected this symbol throughout the manuscript.**

*Line 444*

*Inconsistent depth units: here 4.43m, in Fig A3B.: 4429 mm, would be clearer to unify*

*Line 490*

*As above, 6.29m in the text and 6312mm in Fig. A3D*

**We changed the depths in Figure A3 to a m-scale to keep it consistent with the other mentioned depths. For Figure A3D we implemented the correct breakpoint depth of 6.29 m and replaced it in the line plot.**

*Page 20, Fig. 5*

*Please add markers for hiatus due to technical gap and for LST, similarly to Fig. 6*

**We implemented both markers into Figure 6.**

*Page 22*

*Line 548 – delete double dot*

**Deleted as suggested.**

*Line 549 – please clarify, which transition do you mean here?*

**In this sentence, we refer to both transitions and give more details in the following sentences. We changed the sentence of line 549 as follows: "Moreover, both YD transitions have been predicted within the 95% confidence interval comparable to VT-99 (Table A6) and to the Meerfelder Maar record"**

*Page 37*

*Table A6 is difficult to read in its present format, in particular when reader wants to have a quick glance at some specific numbers. Check the line spacing and names of "events" in the first column. If possible, please add horizontal lines dividing the rows.*

**We agree that the structure needs improvements and reduced the line spacing.**

**References**

**Bonk, A., Müller, D., Ramisch, A., Kramkowski, M. A., Noryśkiewicz, A. M., Sekudewicz, I., Gąsiorowski, M., Luberda-Durnaś, K., Słowiński, M., Schwab, M., Tjallingii, R., Brauer, A., and Błaszkiewicz, M.: Varve microfacies and chronology from a new sediment record of Lake Gościąż (Poland), Quaternary Sci. Rev., 251, 106715, https://doi.org/10.1016/j.quascirev.2020.106715, 2021.**

**Brauer, A., Endres, C., Günter, C., Litt, T., Stebich, M., and Negendank, J. F. W.: High resolution sediment and vegetation responses to Younger Dryas climate change in varved lake sediments from Meerfelder Maar, Germany, Quaternary Sci. Rev., 18, 321–329, https://doi.org/10.1016/S0277-3791(98)00084-5, 1999.**